# GTBENCH: Uncovering the Strategic Reasoning Limitations of LLMs via Game-Theoretic Evaluations

**Jinhao Duan**[1]* **Renming Zhang**[2]* **James Diffenderfer**[3] **Bhavya Kailkhura**[3]
**Lichao Sun**[4] **Elias Stengel-Eskin**[5] **Mohit Bansal**[5] **Tianlong Chen**[5,6,7]† **Kaidi Xu**[1]†

[1]Drexel University [2]Boston University [3]LLNL [4]Lehigh University
[5]UNC Chapel Hill [6]MIT [7]Harvard University

## Abstract

As Large Language Models (LLMs) are integrated into critical real-world applications, their strategic and logical reasoning abilities are increasingly crucial. This paper evaluates LLMs' reasoning abilities in competitive environments through game-theoretic tasks, e.g., board and card games that require pure logic and strategic reasoning to compete with opponents. We first propose GTBENCH, a language-driven environment composing 10 widely-recognized tasks, across a comprehensive game taxonomy: complete versus incomplete information, dynamic versus static, and probabilistic versus deterministic scenarios. Then, we ❶ Characterize the game-theoretic reasoning of LLMs; and ❷ Perform LLM-vs.-LLM competitions as reasoning evaluation. We observe that ❶ LLMs have distinct behaviors regarding various gaming scenarios; for example, LLMs fail in complete and deterministic games yet they are competitive in probabilistic gaming scenarios; ❷ Most open-source LLMs, e.g., CodeLlama-34b-Instruct and Llama-2-70b-chat, are less competitive than commercial LLMs, e.g., GPT-4, in complex games, yet the recently released Llama-3-70b-Instruct makes up for this shortcoming. In addition, code-pretraining greatly benefits strategic reasoning, while advanced reasoning methods such as Chain-of-Thought (CoT) and Tree-of-Thought (ToT) do not always help. We further characterize the game-theoretic properties of LLMs, such as equilibrium and Pareto Efficiency in repeated games. Detailed error profiles are provided for a better understanding of LLMs' behavior. We hope our research provides standardized protocols and serves as a foundation to spur further explorations in the strategic reasoning of LLMs.

## 1 Introduction

Large Language Models (LLMs) are increasingly being integrated into critical real-world applications, such as cybersecurity (Ameri et al., 2021; Aghaei et al., 2022), decision science (Jiang et al., 2023b), and finance (Wu et al., 2023). These areas involve advanced strategic thinking and logical reasoning skills, including the ability to foresee possible dangers and weaknesses (Yao et al., 2024b; Duan et al., 2024a), systematically examine difficulties, and make informed decisions based on provided evidence. However, evaluation environments that thoroughly assess these situations are not sufficiently explored.

There has been an emerging trend where LLMs are evaluated in various interactive role-playing environments, including collaborative environments such as CAMEL (Li et al., 2023), ReConcile (Chen et al., 2023), and competition environments such as Diplomacy (Bakhtin et al., 2022), Werewolf (Xu et al., 2023a), Avalon (Light et al., 2023; Stepputtis et al., 2023), multi-agent debate (Liang et al., 2023; Du et al., 2023; Chan et al., 2023; Xiong et al., 2023), board and card games (Duan et al.,

---

* Equal contribution.
† Correspondence to: Tianlong Chen `tianlong@mit.edu`, Kaidi Xu `kx46@drexel.edu`

38th Conference on Neural Information Processing Systems (NeurIPS 2024).

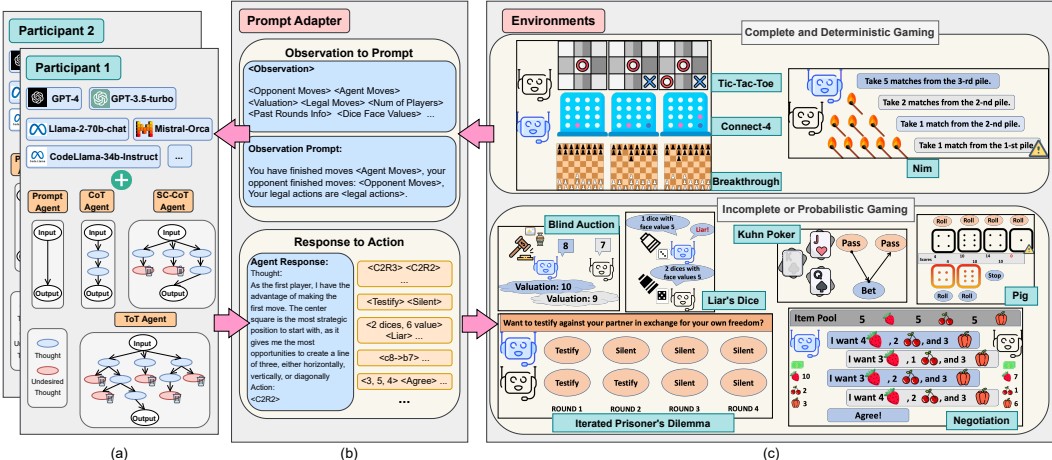

Figure 1: The overall schematic of GTBENCH. There are three main components from *right* to *left*: **Environments** (c) for game hosting, observation providing, and action execution; **Prompt Adapter** (b) for converting observation to prompt and extracting actions from participants' generations; **Participants** (a) for reasoning and action generation.

2024b). By engaging LLMs in simulated scenarios, role-playing-based environments offer useful potential for analyzing the cognitive reasoning abilities of LLMs. However, the extensive background and intricate details involved in role-play-based games dilute the pureness of logic and strategic reasoning that is typically found in game-theoretic tasks. Additionally, the evaluation is primarily verbal as it hinges on spoken or written exchanges between the LLMs. This could mask instances where LLMs might *lack concrete reasoning abilities but navigate the scenario effectively through the proficient use of language*.

***Why are game-theoretic tasks unique and necessary for LLM reasoning evaluation?*** Game-theoretic tasks are typically conceptualized based on prevalent trade-offs and dilemmas manifesting in real-life scenarios and are designed to be easy to understand yet require difficult skills to be mastered. In contrast to the rich narrative contexts afforded in verbal- or role-playing-based games, e.g., Werewolf (Xu et al., 2023a) and Avalon (Light et al., 2023), the reality of game-theoretic games such as Chess and Go involve: ❶ pure logic and strategic reasoning without the added complexity of backgrounds or character roles; ❷ embracing rigorous rules with well-defined action/state space, which allow for an in-depth examination of the strategic reasoning of LLMs.

Hence, in order to spur more research in the LLM Game-Theoretic evaluation domain, we propose GTBENCH, an environment consisting of 10 widely recognized game-theoretic tasks, across a comprehensive taxonomy of games, e.g., complete- (Tic-Tac-Toe, Connect-4, Breakthrough) versus incomplete-information (Kuhn Poker, Liar's Dice) gaming, deterministic (Nim) versus probabilistic (Negotiation, Pig) gaming, static versus dynamic (Iterated Prisoner's Dilemma, Blind Auction) gaming. These environments require a variety of abilities including board strategy, collaboration, auction, and bidding. There are two key issues investigated in this paper:

> **Characterizing Strategic Reasoning of LLMs:** *How LLMs will perform when facing various game-theoretic scenarios? How do they perform compared to conventional solvers? How do essential factors, e.g., pertaining, parameter sizes, and reasoning methods, affect strategic reasoning?*
>
> **LLM-vs.-LLM Competitions as New Reasoning Evaluation:** *A new automated and adaptive benchmark that can be effective in evaluating reasoning errors even for future LLMs.*

To address these crucial problems, we conduct experiments over two configurations: (a) **LLM-vs-Conventional** where conventional solvers such as optimization- or search-based solvers, e.g., Monte-Carlo Tree Search (MCTS) (Chaslot et al., 2008), are taken as the opponent of LLMs; (b) **LLM-vs.-LLM** where two LLMs compete directly to reveal the reasoning limitations in an automated manner. We find that: ❶ LLMs almost always fail when playing against simple MCTS opponents in complete and deterministic gaming scenarios (Section 4.1), while ❷ LLMs remain competitive in incomplete and probabilistic scenarios (Section 4.2); ❸ Code-pretraining benefits game-theoretic

reasoning, e.g., CodeLlama-34b-Instruct (Roziere et al., 2023) achieves comparable results as GPT-3.5-turbo, and significantly outperforms Llama-2-70b-chat (Touvron et al., 2023) (Section 4.3); ❹ Advanced reasoning methods, such as Chain-of-Thought (CoT) (Wei et al., 2022), Self-Consistent CoT (SC-CoT) (Wang et al., 2022b), Tree-of-Thought (ToT) (Yao et al., 2024a) are not always helpful; ❺ Most open-source LLMs are less competitive than commercial LLMs in games with complex rules and large action/state space, while the recently released Llama-3-70b-Instruct (Meta, 2024) makes up for this shortcoming. The interfaces of GTBENCH leaderboard can be found in Appendix A11. Our contributions can be summarized as the following:

- **LLM Game-Theoretic Evaluation (GTBENCH):** An LLM environment supporting 10 well-recognized tasks across comprehensive game-theoretic taxonomy, is presented to spur future work for the community. The code and leaderboard will be public and continuously updated for future reasoning agents and LLMs.

- **Essential Factors for the Strategic Reasoning of LLMs:** We investigate how essential factors, e.g., parameter size, code-pretraining, and reasoning methods, affect strategic reasoning. A detailed error profile is provided for a better understanding of LLMs' behaviors.

- **Characteriz the Game-Theoretic Properties of LLMs:** We characterize distinct LLM behaviors when facing different game-theoretic scenarios, such as LLMs fail in complete-information and deterministic gaming yet remain competitive in probabilistic gaming. We further study the equilibrium and Pareto efficiency during the gameplay.

## 2 Background and Problem Definition

### 2.1 Background and Related Work

**LLM-as-Agent Evaluation.** Several studies have been conducted to measure the effectiveness of LLMs as agents in recent years. Hausknecht et al. (2020) carried out an extensive study to evaluate the performance of LLMs in interactive fiction games. Zhu et al. (2023) provides a valuable dataset for finetuning LLMs to improve usefulness in the strategic game Dungeons & Dragons. GRUE (Ramamurthy et al., 2023) uses reinforcement learning-based metrics to benchmark the performance of generation tasks in six different languages. Gandhi et al. (2023) test the use of LLMs as a broker with human contestants in the negotiation game "Deal or No Deal". A few studies have explored the use of text-based games as a means of facilitating learning in such environments. ALFWorld (Shridhar et al., 2020) introduced a novel virtual environment that allows agents to acquire learning in a text-based environment while executing in a visual environment. The environment was developed in conjunction with Building Understanding in Text world via Language for Embodied Reasoning (BUTLER) agent, which can acquire abstract text knowledge in the text world. Similarly, TextWorld (Côté et al., 2019) is introduced as an environment that enables RL agents to play text games. Wang et al. (2022a) proposed ScienceWorld, a benchmark used for evaluating agents' reasoning ability, and their findings showed that transformer-based models are not effective at reasoning in novel contexts. MTBench (Zheng et al., 2024) introduces LLM-as-a-Judge where GPT-4 (Achiam et al., 2023) is utilized as a judge to evaluate the quality of LLM generations. It indicates that GPT-4 shares close criteria as humans. There have been works evaluating LLMs in solving real-world tasks, such as graph reasoning (Besta et al., 2023), WebShop (Yao et al., 2022), AgentBench (Liu et al., 2023) for pragmatic missions, MINT (Wang et al., 2023b) for tool utilization.

**Multiple LLMs-as-Agents in Gaming.** A key research area is the competition and collaboration between LLMs. Many studies examine LLMs' strategic reasoning and performance, using evaluation frameworks to assess multiple LLM agents in individual games, such as: Social deduction or deception games (Xu et al., 2023a,b; O'Gara, 2023; Light et al., 2023), diplomacy games (Mukobi et al., 2023; , FAIR), negotiation games (Abdelnabi et al., 2023; Davidson et al., 2023), coordination and cooperation games (Akata et al., 2023), and Minecraft (Gong et al., 2023; Wang et al., 2023a; Fan et al., 2022). These works not only provide evaluation frameworks for games and demonstrate the flexibility of LLMs to a variety of gaming tasks but some provide meaningful datasets for fine-tuning, policies for reinforcement learning to produce better strategies, or evaluate the strategic reasoning of LLMs. However, many of these standalone works quantify either individual or a subset of desirable strategic reasoning capabilities of LLMs, such as negotiation, deception, or coordination. Further, they often evaluate these capabilities for LLMs using one or two games which may produce less robust assurances of LLM abilities.

Table 1: Game environments explored in GTBENCH.

| Game | Taxonomy of Games | | | | | Preferred Ability | | | | |
|---|---|---|---|---|---|---|---|---|---|---|
| | Zero-Sum | First-player Advantage | ▲Complete ●Incomplete | ▲Dynamic ●Static | ▲Probabilistic ●Deterministic | Board Strategy | Bids | Collaboration | Bluff | Math |
| Tic-Tac-Toe | ✔ | ✔ | ▲ | ● | ● | ✔ | ✗ | ✗ | ✗ | ✗ |
| Connect-4 | ✔ | ✔ | ▲ | ● | ● | ✔ | ✗ | ✗ | ✔ | ✔ |
| Kuhn Poker | ✔ | ✔ | ● | ● | ▲ | ✗ | ✗ | ✗ | ✔ | ✗ |
| Breakthrough | ✔ | ✗† | ▲ | ● | ● | ✔ | ✗ | ✗ | ✗ | ✗ |
| Liar's Dice | ✔ | ✗ | ● | ● | ▲ | ✗ | ✔ | ✗ | ✔ | ✔ |
| Blind Auction | ✗ | ✗ | ● | ▲ | ▲ | ✗ | ✔ | ✗ | ✗ | ✔ |
| Negotiation | ✗ | ✗ | ● | ● | ▲ | ✗ | ✗ | ✔ | ✔ | ✔ |
| Nim | ✔ | ✔ | ▲ | ● | ● | ✗ | ✗ | ✗ | ✗ | ✔ |
| Pig | ✗ | ✗ | ▲ | ● | ▲ | ✗ | ✗ | ✗ | ✗ | ✗ |
| Iterated Prisoner's Dilemma | ✗ | ✗ | ▲ | ▲ | ● | ✗ | ✗ | ✔‡ | ✗ | ✔ |

† : Breakthrough has a slight first-player advantage which is not as significant as others.
‡ : The iterated version of Prisoner's Dilemma allows participants access to the actions made by their opponents in the past rounds, achieving implicit collaboration.
†† : Inapplicable due to complex combination and dynamic environment.

We make an additional crucial contribution in this line of work by measuring strategic reasoning capabilities with games that are not found in the existing unified benchmark suites (Zhang et al., 2024), such as clembench (Chalamalasetti et al., 2023) focusing on conversational agents over non-zero-sum games and LMRL-Gym (Abdulhai et al., 2023) on verbal reinforcement learning tasks. (Chen et al., 2024) and Duan et al. (2024b) also proposes multi-agent strategic reasoning evaluation. However, they overlooked the analysis of LLM behaviors in response to different game-theoretic scenarios and their associated properties. Differently, GTBENCH seeks to provide a unified suite of games that are carefully curated to (1) evaluate a comprehensive collection of strategic reasoning abilities for a given agent and (2) enable competition-based scenarios (i.e., LLM agent-1 vs LLM agent-2) allowing for competition-based comparisons of strategic reasoning capabilities by LLM-based agents.

## 2.2 Problem Definition

**Notation: Gameplay.** We formulate the gameplay as a Markov Decision Process $(\mathcal{S}, \mathcal{A}, \mathcal{M}, \mathcal{O})$ under a given game environment, among the alternating interaction of two participants. This process composes of an infinite state space $\mathcal{S}$, an infinite action space $\mathcal{A}$, the participants $\mathcal{M} = \{\mathcal{M}_1, \mathcal{M}_2\}$, and an observation space $\mathcal{O}$. Considering the decision of $\mathcal{M}_i$ $(i = 1, 2)$ at the $t$-th step of the process, we denote by $s_t \in \mathcal{S}$ the state that $\mathcal{M}_i$ are placed and $o_t \in \mathcal{O}$ the observation that $\mathcal{M}_i$ are observing. We assume $\mathcal{M}_i$ follows policy $\pi_{\theta_i}(a_t|s_t, o_t)$ for state transition $\mathcal{T} : \mathcal{S} \times \mathcal{A} \to \mathcal{S}$, where $a_t \in \mathcal{A}$ is the action sampled by $\pi_{\theta_i}$ under conditions $s_t$ and $o_t$. $\theta_i$ is determined by the implementation by $\mathcal{M}_i$, e.g., optimization-based solver, LLM-driven agents, which will be discussed in Section 3.2 in detail. In this way, the two-participate gameplay can be represented as $(s_0, a_0, s_1, a_1, s_2, \cdots, s_n)$, where $s_0$ is the initial state and $s_n$ is a terminal state, i.e., end of the game. The progress is driven by the alternating execution of actions sampled by participants. Please refer to Section 3.1 and Appendix A2 for all the supported games with the corresponding actions and observations.

**Evaluation Metric: Normalized Relative Advantage.** We introduce **Normalized Relative Advantage (NRA)**, denoted $NRA(\mathcal{M}_i, \mathcal{M}_o, f_s)$, to measure to relative advantage of $\mathcal{M}_i$ when competing against $\mathcal{M}_o$, under the score calculation $f_s$:

$$NRA(\mathcal{M}_i, \mathcal{M}_o, f_s) = \frac{\sum_m f_s(\mathcal{M}_i, m) - \sum_m f_s(\mathcal{M}_o, m)}{\sum_m f_s(\mathcal{M}_i, m) + \sum_m f_s(\mathcal{M}_o, m)},$$

where $f_s(\mathcal{M}_i, m)$ refers to the score earned by $\mathcal{M}_i$ at the $m$-th match ($1 \leq m \leq K$, $K$ is the number of performed matches):

- For zero-sum games, e.g., Tic-Tac-Toe,

$$f_s(M_i, m) = \begin{cases} 1, & \text{if } \mathcal{M}_i \text{ wins at the } m\text{-th match} \\ 0, & \text{if } \mathcal{M}_i \text{ loses at the } m\text{-th match} \\ 0.5, & \text{if } \mathcal{M}_i \text{ and } \mathcal{M}_o \text{ achieve a draw} \end{cases}$$

- For non-zero-sum games, e.g., Blind Auction, $f_s(M_i, m)$ is the rewards earned by $\mathcal{M}_i$ at the $m$-th match.

$NRA(\mathcal{M}_i, \mathcal{M}_o, f_s)$ is naturally normalized to $[-1, 1]$, providing an interpretable meaning regarding the performance of $\mathcal{M}_i$: $NRA(\mathcal{M}_i, \mathcal{M}_o, f_s) > 0$ means $\mathcal{M}_i$ is better than $\mathcal{M}_o$; $NRA(\mathcal{M}_i, \mathcal{M}_o, f_s) < 0$ means $\mathcal{M}_i$ is worse than $\mathcal{M}_o$; $NRA(\mathcal{M}_i, \mathcal{M}_o, f_s) = 0$ means $\mathcal{M}_i$ is as competitive as $\mathcal{M}_o$.

**Evaluation Metric: Elo Rating.** Following the conventional rating mechanism in the real world, e.g., Chess, we employ the popular **Elo Rating** (Elo, 1960) for calculating the relative skill levels of players in zero-sum games. Please refer to Appendix A7 for more details of Elo rating.

## 3 GTBENCH: Game-Theoretic Evaluation of LLMs

GTBENCH is a language-driven environment, making participating agents compete against each other in a game-theoretic manner. It is designed to be flexible and extensible, providing unified interfaces to participants and games, and supporting various multi-turn-based games which can be extended in the future. The overall framework is presented in Figure 1. There are three main components: *Environment*, *Prompt Adapter*, and *Participant*. Please refer to Appendix A1 for a detailed introduction of each component.

### 3.1 Taxonomy of Game-Theoretic Tasks

The chosen tasks and their detailed configurations are presented in Table 1. To comply with the common taxonomy (Lanctot et al., 2019) of game-theoretic tasks and provide diverse gaming scenarios, GTBENCH supports 10 different gaming environments, including `Tic-Tac-Toe`, `Connect-4`, `Kuhn Poker`, `Breakthrough`, `Liar's Dice`, `Blind Auction`, `Negotiation`, `Nim`, `Pig`, `Iterated Prisoner's Dilemma`, covering 6 mainstream game-theoretic configurations, including *complete-* and *incomplete-information* gaming, *dynamic* and *static* gaming, and *probabilistic* and *deterministic* gaming. The preferred abilities of each game could be characterized as the combination of *board strategy*, *bids*, *collaboration*, *bluff*, and *math*. Please refer to Appendix A2.1 for the rules of each game and Appendix A2.2 for an explanation of game-theoretic taxonomy.

### 3.2 Participants and Protocols

**Conventional Agents** output actions through a conventional optimization or searching process. To provide fair comparisons, we employ the powerful Monte Carol Tree Search (MCTS) (Chaslot et al., 2008) as the conventional agent for most of the games, with the number of simulations as 1000. Since `Iterated Prisoner's Dilemma` is dynamic gaming with very limited action space, i.e., <TESTIFY> or <SILENT>, we utilize the more popular Tit-for-Tat (Axelrod, 1981) strategy, which simply repeating the opponent's last action, as the conventional agent. We also include Random Agent that randomly selects action at each turn, serving as a baseline and sanity check. Please refer to Appendix A3.1 for more details about MCTS Agent and Tit-for-Tat Agent.

**LLM-Driven Reasoning Agent** consists of backbone LLMs and reasoning paradigms. For reasoning schemes, we consider the following reasoning paradigms as they are widely known to be effective for general reasoning tasks: ❶ *Prompt*: Directly Prompt LLMs to generate responses, without additional reasoning steps; ❷ *Chain-of-Thought (CoT)* (Wei et al., 2022): CoT Agent prompts LLMs by thinking step by step; ❸ *Self-Consistent CoT* (Wang et al., 2022b): SC-CoT Agent prompts LLMs by generating multiple step-by-step thinking trajectories and performing majority voting to get the final response. The number of trajectories is set to 5 in this paper; ❹ *Tree-of-Thought (ToT)* (Yao et al., 2024a): ToT Agent prompts LLMs to generate responses by incorporating exploration and deliberate decision-making, e.g., self-evaluation. The number of sequences for both answer generation and answer evaluations is set to 3.

**Prompt Templates.** Prompts are designed to be modular, consisting of four individual components: *System Prompt*, *Head Prompt*, *Observation Prompt*, and *Reasoning Prompt*. Reasoning prompts, e.g., CoT/ToT, are designed to only focus on instructing LLM how to think, regardless of the game environment. Thus, they could be automatically adapted when adding a new game. Please refer to Appendix A5 for the detailed prompts and observations for each game and agent.

**Sanity Check.** We provide the task completion rates of all the LLMs and reasoning agents in Appendix A5.6. We show that all the LLM agents achieve $\geq 90\%$ completion rate, indicating that the prompts are properly configured and LLMs are capable of following instructions to finish the game.

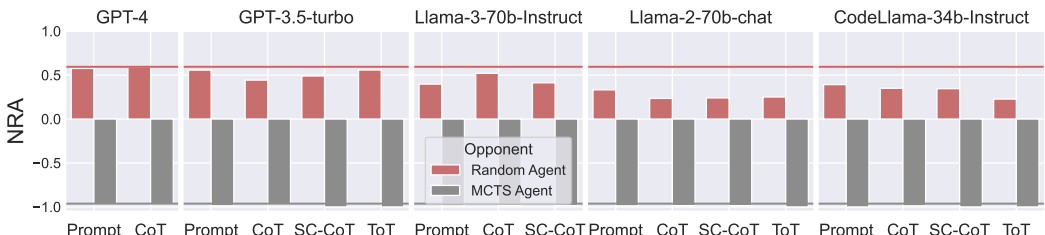

Figure 2: The NRA of state-of-the-art LLM-driven reasoning agents when against MCTS Agents and Random Agents, over complete and deterministic scenarios. Red and gray lines mean the maximum NRA achieved by LLM agents.

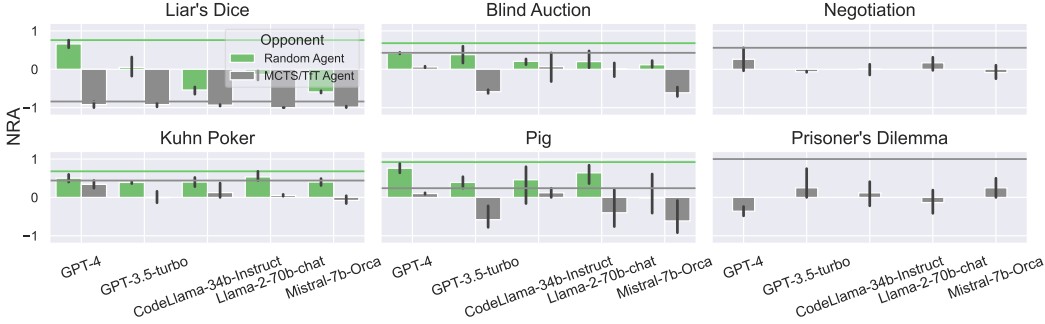

Figure 3: The game-wise NRA of LLMs when against MCTS/TfT Agents and Random Agents, over incomplete and probabilistic scenarios. Error bars are obtained over different reasoning methods. Green and gray lines mean the maximum NRA achieved by LLM agents.

# 4   Are LLMs Capable of Strategic Reasoning?

In this section, we evaluate the strategic reasoning capabilities of LLMs by conducting experiments among conventional solvers and LLM-driven agents.

**Experimental Settings.** We consider well-recognized LLMs such as commercial LLMs: GPT-3.5-turbo-1106 and GPT-4-0613 (Achiam et al., 2023), and open-source LLMs: Llama-3-70b-Instruct (Meta, 2024), Deepseek-LLM-67b-chat (Bi et al., 2024), Llama-2-70b-chat (Touvron et al., 2023), CodeLlama (Roziere et al., 2023), and Mistral-7b-Orca (Jiang et al., 2023a; Mukherjee et al., 2023). For all the LLMs, the temperature is set to 0.2 and the max number of generated tokens is 1024. For each competition, we run 50 valid matches. The final performance is measured by the averaged NRA over the 50 valid matches. To mitigate the first-player advantage, we have each participant take the first turn in 25 matches.

## 4.1   Complete and Deterministic Gaming

There are four complete and deterministic tasks supported in GTBENCH: `Tic-Tac-Toe`, `Connect-4`, `Breakthrough`, and `Nim`. We compare LLM-driven agents with Random Agent and MCTS Agent. Results are summarized in Figure 2. In general, we show that all LLMs achieve substantial relative advantages when competing against the Random Agent. Among all the agents, GPT-4 w/ CoT reasoning achieves the highest NRA. For open-source LLMs, Llama-3-70b-Instruct outperforms other open-source LLMs, achieving comparable capabilities as GPT-4.

However, when competing against the MCTS Agent, all the LLM agents equipped with various reasoning methods achieve NRA as $-1$, meaning that LLM agents can barely win even a single match. This is because for board games with moderate action/state space such as the four involved complete and deterministic games in GTBENCH, MCTS agents with a sufficient number of simulations can achieve near-optimal strategies. Consequently, LLMs are not competitive in complete and deterministic games.

## 4.2   Probabilistic and Dynamic Gaming

There are five probabilistic game-theoretic gaming tasks: `Kuhn Poker`, `Liar's Dice`, `Blind Auction`, `Negotiation`, `Pig`, and one dynamic task: `Iterated Prisoner's Dilemma`. We group these games together as they all involve stochasticity in the gameplay, which is essentially different from complete and deterministic games. The Random Agent as the opponent is omitted for both Negotiation and Iterated Prisoner's Dilemma because the Random Agent rarely chooses to collaborate, resulting

Table 2: Code-pretraining benefits strategic reasoning. Gray rows are code-pretrained LLMs.

| Model | avg. NRA in Det. Games | avg. NRA in Prob. | avg. NRA |
|---|---|---|---|
| GPT-4 | 0.09 | 0.15 | 0.13 |
| Llama-3-70b-Instruct | -0.07 | 0.11 | 0.04 |
| Llama-2-70b-chat | -0.25 | -0.17 | -0.20 |
| CodeLlama-34b-Instruct | **-0.05** | **0.02** | **-0.01** |
| Deepseek-LLM-7b-chat | -0.09 | -0.08 | -0.08 |
| Deepseek-LLM-67b-chat | **0.10** | -0.17 | -0.05 |
| Deepseek-Coder-6.7b-instruct | -0.14 | **0.07** | **-0.03** |

in meaningless evaluation. Results are summarized in Figure 3. When competing against the MCTS Agent, it is shown that `Liar's Dice` shares a similar trend as the complete and deterministic scenarios (Figure 2), where LLM-driven agents achieve near $-1$ NRA. This is because the 2-player `Liar's Dice` has very limited stochasticity, making the gameplay tend to be complete information. For other tasks, we found that LLMs do not always fail. We observe that the NRA of LLM agents is close to 0 over all the tasks, indicating that they are equally competitive as conventional solvers or even better (e.g., `Kuhn Poker` where GPT-4 outperforms MCTS Agent).

## 4.3 LLM-vs.-LLM Competition

We investigate whether popular LLMs remain competitive in game-theoretic scenarios. Specifically, we take GPT-3.5-turbo with Prompt Agent as the *common opponent* and make other LLM-driven agents compete against it. Please refer to Figure A6 for the full leaderboard evaluated by NRA. The Elo rating results are placed in Table 6. In general, GPT-4 is the most powerful LLM in strategic reasoning among all the examined LLMs. Moreover, Llama-3-70b-Instruct achieves comparable performances as GPT-4 and outperforms GPT-3.5-turbo. Here we break the results into 3 takeaways:

**Code-Pretraining Benefits Game-Theoretic Tasks.** In Table 2, we show code-pretrained LLMs, e.g., CodeLlama-34b-Instruct and Deepseek-Coder-6.7b-Instruct, significantly outperform larger chat LLMs, e.g., Llama-2-70b-chat and Deepseek-LLM-67b-chat. These code-pretrained LLMs have less than half of the parameters, suggesting that code-pretraining benefits game-theoretic tasks. This verifies recent discoveries where code-pretraining benefits logical reasoning (Madaan et al., 2022; Liang et al., 2022; Ma et al., 2023).

**Advanced Reasoning Methods Do Not Always Help.** We observe that advanced reasoning methods may lead to worse results in game-theoretic scenarios. To make it more clear, we present the averaged NRA obtained by reasoning methods across different LLMs when against Random Agent in Figure 4. In general, only Mistral-7b-Orca has a substantial improvement when equipped with CoT reasoning while advanced reasoning leads to worse results for other LLMs.

In Table 3, we present the results when against GPT-3.5-turbo w/ Prompt Agent. We show that advanced reasoning benefits powerful LLMs, e.g., GPT-3.5-turbo, while it results in worse results for other LLMs. It suggests that advanced reasoning is a

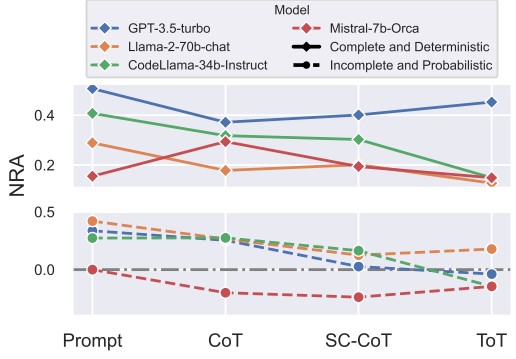

Figure 4: The NRA of LLM agents when competing against Random Agent. Advanced reasoning does not always result in better results.

Table 3: The NRA of LLM agents w/ CoT reasoning. Cyan cells mean CoT results in better performance. Magenta cells mean CoT results in worse performance.

| Opponent | Model | Reasoning | avg. NRA ↑ |
|---|---|---|---|
| GPT-3.5-turbo w/ Prompt Agent | GPT-3.5-turbo | Prompt | 0.00 |
| | | CoT | 0.02 |
| | Llama-3-70b-Instruct | Prompt | 0.04 |
| | | CoT | 0.07 |
| | GPT-4 | Prompt | 0.13 |
| | | CoT | 0.13 |
| | CodeLlama-34b-Instruct | Prompt | -0.01 |
| | | CoT | -0.09 |
| GPT-4 w/ Prompt Agent | CodeLlama-34b-Instruct | Prompt | -0.01 |
| | | CoT | -0.04 |
| | Llama-2-70b-chat | Prompt | -0.10 |
| | | CoT | -0.23 |

double-edged sword: ❶ powerful LLMs are capable of leveraging advanced reasoning to achieve better results; ❷ advanced reasoning may also impose reasoning errors and risks during the inference of ordinary LLMs. In Appendix A8, we further examine five different CoT strategies over the GPT-3.5-turbo model to mitigate the effect brought by prompt sensitivity, along with some failure cases presented. These CoT prompts resulting in different performances are all worse than the naive Prompt Agent.

**Most Open-source LLMs are Less Competitive than Commercial LLMs in Complex Games.** We observe that most of open-source LLMs such as Llama-2-70b-chat and CodeLlama-34b-Instruct are not good at games with complex rules and board states. In Table 4, we present the average NRA when including and excluding `Breakthrough`[3]. It is shown that both Llama-2-70b-chat and CodeLlama-34b-Instruct fail in `Breakthrough`, resulting in worse NRA scores than GPT-4. However, we found that the recently released Llama-3-70b-Instruct (Meta, 2024) has a significant performance in `Breakthrough`. This indicates that open-source LLMs achieve comparable capabilities when dealing with complex tasks and environments as commercial LLMs.

Table 4: The average NRA of LLM-driven agents when `Breakthrough` is included and excluded.

| Taxonomy | GPT-4 | Llama-3-70b-Instruct | CodeLlama-34b-Instruct | Llama-2-70b-chat |
|---|---|---|---|---|
| **w** `Breakthrough` | 0.13 | 0.04 | -0.01 | -0.20 |
| **w/o** `Breakthrough` | 0.11 (-0.02) | -0.01 (-0.05) | 0.08 (+0.09) | -0.18 (+0.02) |

## 4.4 Error Profiles

We introduce the most prevalent mistake patterns observed across different games, comprising *Misinterpretation*, *Factual Inaccuracies*, *Overconfidence*, *Calculation Mistakes*, and *Endgame*:

**Misinterpretation** denotes the misinterpretation of the game's current state by LLMs, including errors like misattributing piece ownership and failing to recognize vacant spots on the board. **Factual Errors** refer to situations where the

Table 5: Quantitative results of error patterns.

| Model | Percentage of Error Patterns (%) | | | | |
|---|---|---|---|---|---|
| | Endgame Misdetection | Mis-interpretation | Over-confidence | Calculation Error | Factual Error |
| GPT-4 | 33.33 | 9.80 | 15.69 | 9.80 | 45.10 |

player has a reasonable plan but their actions do not align with their plan. For instance, in `Breakthrough`, GPT-4 w/ CoT agent plans to fend off frontal attacks by the opponent, which is reasonable. However, it takes rear pieces to achieve that, which is impossible. **Over-confidence** describes a scenario where a player overlooks potential risks in pursuit of greater rewards. **Calculation Errors** refer to errors that occur in arithmetic, such as calculating XOR in `Nim`. **Endgame Misdetection** means a failure to recognize immediate win/lose situations, e.g., a player fails to recognize a potential winning move. Demonstrations of each mistake pattern are presented in Appendix A9.

In Table 5, we present the quantitative results regarding these error patterns. It is obtained from GPT-4 w/ CoT agent when playing against conventional solvers, e.g., MCTS/TfT agent, as the opponent. We manually examined a total of 157 turns (50 matches, with 5 turns per match). We observe that LLM agents are capable of generating reasonable planning/strategies. However, they have difficulties in selecting the correct actions to align with their thoughts. Also, LLMs miss endgame situations, leading to a failure to recognize winning and losing moves.

Table 6: The Elo rating results of LLM-vs.-LLM experiments.

| Model | Tic-Tac-Toe | Breakthrough | Blind Auction | Kuhn Poker | Liar's Dice | *avg.* Elo |
|---|---|---|---|---|---|---|
| GPT-4 | 1554.34 | 1667.11 | **1581.94** | 1479.87 | 1676.70 | **1591.99** |
| Llama-3-70b-Instruct | 1371.68 | **1669.42** | 1524.11 | **1625.46** | **1694.64** | 1577.06 |
| GPT-3.5-turbo | 1579.80 | 1576.37 | 1514.27 | 1441.80 | 1459.26 | 1514.30 |
| CodeLlama-34b-Instruct | **1589.94** | 1398.10 | 1533.48 | 1414.57 | 1374.40 | 1462.10 |
| Llama-2-70b-chat | 1479.08 | 1320.42 | 1484.32 | 1521.82 | 1485.00 | 1458.13 |
| Mistral-7B-Instruct | 1440.15 | 1338.57 | 1361.89 | 1516.48 | 1310.00 | 1393.42 |

---

[3]`Breakthrough` has larger action/state space than other complete-information games.

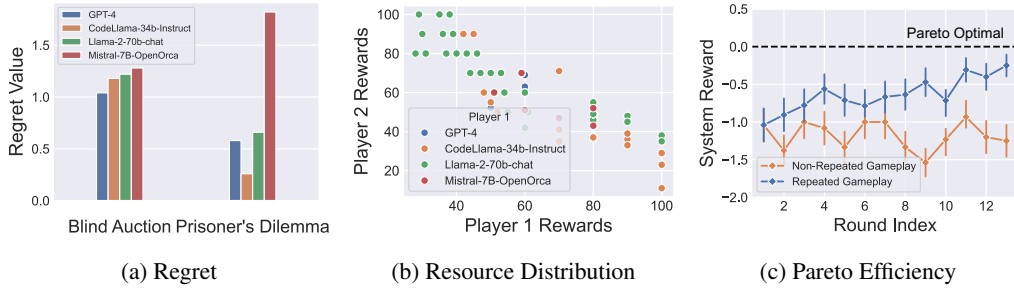

| (a) Regret | (b) Resource Distribution | (c) Pareto Efficiency |

Figure 5: Game-theoretic properties. The results are obtained when competing against GPT-3.5-turbo w/ Prompt Agent as the opponent. In (b), each dot $(x, y)$ represents an agreement in a resource distribution with Player 1 obtaining reward $x$ and Player 2 obtaining reward $y$. In (c), the system reward is calculated by the sum of the payoffs of all players.

## 5 The Game-Theoretic Properties of LLMs

**Nash Equilibrium with Regret.** In game theory, being close to a Nash Equilibrium (Nash Jr, 1950) indicates that the strategies chosen by the players are near to optimal. It has been popular to approximate Nash Equilibrium with Regret[4] (Johanson et al., 2012; Nisan and Noti, 2017; Zinkevich et al., 2007). In Figure 5a, we present the regret values of LLMs on `Blind Auction` and `Iterated Prisoner's Dilemma`. Please refer to Appendix A10 for how regret values are calculated for these two tasks. For `Blind Auction`, GPT-4 shows lower Regret, indicating achieving closer to optimal solutions than other LLMs. However, in `Iterated Prisoner's Dilemma`, CodeLlama-34b-Instruct exhibits lower regret compared to GPT-4. Through human examination, we found that this is because GPT-4 tends to <Silent> more frequently, whereas Codellama has a significantly higher probability of <Testify>. This discrepancy may be due to the human preference alignment in GPT-4, such as a higher emphasis on morality (Pan et al., 2023) or maximizing system reward[5], which makes GPT-4 less likely to <Testify>.

**Pareto Efficiency.** We study Pareto Efficiency in two games: `Negotiation` and `(Iterated) Prisoner's Dilemma`. In Figure 5b, we count all agreements reached by participants and record the values attributed to each based on the agreed division. Most agreements result in substantial values for both participants, though some LLMs, like Llama-2-70b-chat and CodeLlama-34b-Instruct, may accept unfair resource divisions. In contrast, GPT-4 and Mistral struggle to reach agreements and tend to negotiate for Pareto improvements. A repeated game is a standard game that is played multiple times by the same players, with each player is able to observe the history of past plays (Aumann et al., 1995; Akata et al., 2023). In Figure 5c, we investigate the Pareto Improvement in `Iterated Prisoner's Dilemma` and ordinary `Prisoner's Dilemma`, i.e., each round is played individually. The Pareto Improvement is observed in the repeated-game scenario during the rounds, indicating that LLMs are capable of leveraging history to adjust their strategies.

## 6 Conclusion

This work investigated LLMs' strategic and logical reasoning abilities under competitive scenarios. To achieve this, we created a broad evaluation scope by considering various classic and LLM-based gaming agents and 10 representative games. We conducted the benchmark study of game-theoretic evaluations for LLMs, shedding light on their reasoning performance. Our extensive evaluations revealed insightful LLMs' gaming behavior, such as their intrinsic failure in complete and deterministic games, impressive reasoning in incomplete and probabilistic games, and benefiting from code-generation pertaining and appropriate prompt designs.

**Limitations** This research prompts LLMs to generate actions regarding various game scenarios, relying on pre-defined prompt templates. Thus, the results may suffer from certain variances introduced by prompt sensitivities. Although the introduced games are popular, their actions/state

---

[4]Regret (Zinkevich et al., 2007) measures how much a player would have improved their outcome by choosing a different strategy, given what they know now after the game has played out.

[5]<Silent> maximizes the system reward in `Iterated Prisoner's Dilemma`

space is limited, which may not be well-distinguished for LLMs in the same skill levels. The generated actions may be illegal due to the incapabilities of the following instructions.

**Impact Statements**  This paper examines the game-theoretic task proficiency of AI models. We acknowledge concerns about models becoming autonomous entities with their own objectives, especially in deception or negotiation scenarios. It's important to note that our research measures the current capabilities of models, rather than enhancing their abilities. We do not train AI models to be competent in game theory tasks or to bluff or defect. Instead, we assess existing competencies, contributing to a deeper understanding that can inform innovative measures against potential risks. We believe our work paves the way for responsible and effective AI safety.

## Acknowledgement

This work was performed under the auspices of the U.S. Department of Energy by the Lawrence Livermore National Laboratory under Contract No. DE- AC52-07NA27344 and was supported by the LLNL LDRD Program under Project No. 23-ERD-030 and 24-ERD-058. This work was partially supported by the NSF award FMitF-2319242. It was also partially supported by NSF-AI Engage Institute DRL-2112635 and DARPA MCS Grant N66001-19-2-4031. The views, opinions, and/or findings contained in this article are those of the authors and not of the funding agency.

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

# Appendix

## A1 Overall Architecture

There are three main components in GTBENCH:

- **Environment.** The environment ( Figure 1 (c)) is responsible for overseeing the crucial processes related to gameplay. Specifically, it is tasked with building up observations, managing gameplay, and applying the actions obtained from participants. In this paper, all of the gaming environments are built on top of OpenSpiel Lanctot et al. (2019).

- **Prompt Adapter.** The prompt adapter ( Figure 1 (b)) plays a vital role in facilitating effective communication between the environment and the virtual participants. It serves as an intermediary between the two entities by receiving observations from the environment, which it then translates into unified observation prompts. The prompts are then parsed and sent to the participating agents to formulate their responses. The adapter is also responsible for obtaining actions from the participants, which it transforms into legal actions before parsing them to the environment for game execution.

- **Participant.** The participants ( Figure 1 (a)) involved in the gaming process generate responses according to the observation prompts received from the Prompt Adapter. These responses consist of actions that participants intend to take in this turn.

## A2 Gameplay Configurations

### A2.1 Games Introduction

**Tic-Tac-Toe**[6] is a paper-and-pencil game for two players who take turns marking the spaces in a three-by-three grid with X or O. The player who succeeds in placing three of their marks in a horizontal, vertical, or diagonal row is the winner. It is a solved game, with a forced draw assuming optimal play from both players.

- **Observation (input):** Our observation contains "opponent moves" and "self moves". "Opponent moves" contains all the current opponent agent's historical actions. "Self moves" contains all the current agent's history actions.

- **Actions:** We define our action in the following format: $<\text{C}x\text{R}y>$, in which C and R mean columns and rows respectively, while $x$ and $y$ mean the index of column and row. Each player may make their own action in turn.

**Prisoner's Dilemma**[7] is a game theory thought experiment that involves two rational agents, each of whom can cooperate for mutual benefit or betray their partner ("defect") for individual reward.

- **Observation (input):** Our observation contains "opponent moves" and "self moves". "Opponent moves" contains all the current opponent agent's historical actions. "Self moves" contains all the current agent's history actions.

- **Actions:** We define our action in the following format: $<\text{Silent}>$ or $<\text{Testify}>$. All players must take their action simultaneously.

**Breakthrough**[8] Breakthrough is an abstract strategy board game invented by Dan Troyka in 2000 and made available as a Zillions of Games file (ZRF). It won the 2001 8x8 Game Design Competition. The first player to reach the opponent's home row — the one farthest from the player — is the winner. In our work, we scale the size of the board to 3*8 while maintaining its competitiveness.

- **Observation (input):** Our observation contains "opponent moves", "self moves", and "board preview". "Opponent moves" contains all the current opponent agent's historical actions. "Self moves" contains all the current agent's history actions. The "board preview" feature maintains the status of each grid on the board through a list of strings, denoting whether it contains a black piece, a white piece, or is empty.

---

[6] https://en.wikipedia.org/wiki/Tic-tac-toe
[7] https://en.wikipedia.org/wiki/Prisoner%27s_dilemma
[8] https://en.wikipedia.org/wiki/Breakthrough_(board_game)

- **Actions:** We define our action in the following format: A$x$->B$y$, in which A and B mean the current column index and destination column index respectively, while $x$ and $y$ mean the index of current row and destination row. Each player may make their own action in turn.

**Connect Four**[9] is a game in which the players choose a color and then take turns dropping colored tokens into a six-row, seven-column vertically suspended grid. The pieces fall straight down, occupying the lowest available space within the column. The objective of the game is to be the first to form a horizontal, vertical, or diagonal line of four of one's own tokens.

- **Observation (input):** Our observation contains "opponent moves" and "self moves". "Opponent moves" contains all the current opponent agent's historical actions. "Self moves" contains all the current agent's history actions.
- **Actions:** We define our action in the following format: $<Cx>$ in which C means column, while $x$ means the index of column. Each player may make their action in turn.

**Blind Auction**[10] is a common type of auction. In this type of auction, all bidders simultaneously submit sealed bids so that no bidder knows the bid of any other participant. The highest bidder pays the price that was submitted. All players must take their action simultaneously.

- **Observation (input):** Our observation contains "valuation"."Valuation" contains each of the values of all the items for the current player.
- **Actions:** We define our action in the following format: $<x>$, in which $x$ represents the amount that a certain player would like to bid for.

**Kuhn Poker**[11] is a simplified form of poker. Kuhn is a simple model zero-sum two-player imperfect-information game, amenable to a complete game-theoretic analysis. In Kuhn poker, the deck includes only three playing cards, for example, a King, Queen, and Jack. One card is dealt to each player, which may place bets similarly to a standard poker. If both players bet or both players pass, the player with the higher card wins, otherwise, the betting player wins.

- **Observation (input):** Our observation contains "card" , and "moves". Among these, "card" denotes the current player's hand card in this match, while "moves" represents the history of all characters' moves together with the index of the rounds.
- **Actions:** We define our action in the following format: $<Pass>$ or $<Bet>$. Each player may make their own action in turn.

**Liar's Dice**[12] is a class of dice games for two or more players requiring the ability to deceive and detect an opponent's deception.

- **Observation (input):** Our observation contains: "Self dice face value" and "last move". "Self dice face value" describes all the face values of dices the current player has, while "last move" represents the previous player's action.
- **Actions:** We define our action in the following format: $< x$ dices, $y$ value$>$ or $<Liar>$. Among these, $x$ means the quantity of dice, and $y$ means the face values of the dice. The option "Liar" denotes the current player wants to stop and challenge the previous players. Each player may make their own action in turn.

**Pig**[13] is a simple dice game. Players take turns to roll a single dice as many times as they wish, adding all roll results to a running total, but losing their gained score for the turn if they roll a 1.

- **Observation (input):** Our observation contains: "self current score", "opponent current score", and "turn total score". "Self current score" and "opponent current score" represent the game culminated score of the current player and opponent player respectively. While "turn total score" denotes the sum of the score of the current turn.

[9] https://en.wikipedia.org/wiki/Connect_Four
[10] https://en.wikipedia.org/wiki/First-price_sealed-bid_auction
[11] https://en.wikipedia.org/wiki/Kuhn_poker
[12] https://en.wikipedia.org/wiki/Liar%27s_dice
[13] https://en.wikipedia.org/wiki/Pig_(dice_game)

- **Actions:** We define our action in the following format: $<$stop$>$ or $<$roll$>$. Each player may make their own action in turn.

**Nim**[14] is a mathematical game of strategy in which two players take turns removing objects from distinct heaps or piles. On each turn, a player must remove at least one object and may remove any number of objects provided they all come from the same heap or pile.

- **Observation (input):** Our observation contains: "piles". "Piles" denotes the number of matches different piles have.
- **Actions:** We define our action in the following format: $<$pile:$x$, take:$y>$. Among these, $x$ represents the index of the pile that the current player takes, and $y$ represents the number of matches the current player takes. Each player may make their own action in turn.

**Negotiation**[15]

- **Observation (input):** Our observation contains: "turn type", "item pool", "most recent proposal", "most recent utterance", and "self value vector". "turn type" is an enum variable, it has two options: proposal and utterance. The "Proposal" is the turn that the current player could think about the desired quantities of the items, and the "Utterance" is the turn that the current player states the values to its opponent. "item pool" represents the quantities of all the items."most recent proposal" and "most recent utterance" represent the opponent's latest proposal and utterance. "self value vector" represents how much the value of the items to the current player.
- **Actions:** We define our action in the following format: $<$Agree$>$ or $< x, y, z >$. Among these, $<$Agree$>$ represents the current player agreeing on the opponent's utterance. $x$, $y$, and $z$ represent the quantities of different items that the current player wants to get.

## A2.2 Gaming-Theoretic Taxonomy

**Complete and Incomplete Information** One fundamental dimension along which games are classified is the level of information available to players. In *complete information* games, players possess perfect knowledge regarding the game's structure, including the available strategies, payoffs, and the actions taken by other players. Examples of complete information games include canonical examples like chess and `Tic-Tac-Toe`, where all relevant information is transparent to all players throughout the game. Conversely, *incomplete information* games involve situations where players must make decisions without having full knowledge of the game's parameters or the actions of other players. Classic examples of incomplete information games include strategic interactions in economics, such as auctions or negotiations, where players have limited knowledge about the valuations or preferences of other participants.

**Dynamic and Static** Another crucial dimension for classifying games is the timing of players' decisions. In *static games*, players make decisions simultaneously, without the opportunity to observe or react to other players' moves. Examples of static games include simultaneous-move games like the `Iterated Prisoner's Dilemma`. In contrast, *dynamic games* involve sequential decision-making, where players observe previous moves before choosing their actions. Dynamic games encompass a wide range of strategic environments, from turn-based board games like chess to dynamic settings like `Kuhn Poker`, where players strategically make their actions based on the unfolding dynamics of the game.

**Probabilistic and Deterministic** Games can also be differentiated based on the role of uncertainty in decision-making. In *deterministic games*, the outcomes of players' actions are fully determined by the game's rules and the strategies chosen by players. Deterministic games include classic examples like chess or `Tic-Tac-Toe`, where each move leads to a predictable outcome based on the game's rules and the players' strategies. Conversely, *probabilistic games* involve randomness or uncertainty in determining outcomes. This uncertainty can stem from elements such as dice rolls, card draws. Examples of probabilistic games include games of chance like `Kuhn Poker`, `Liar's Dice`, or `Pig`, where players must contend with the inherent uncertainty of probabilistic outcomes.

---

[14]https://en.wikipedia.org/wiki/Nim
[15]https://arxiv.org/pdf/1706.05125.pdf

## A3 Participants

### A3.1 Conventional Agent

**MCTS** Chaslot et al. (2008) is a heuristic search algorithm that has gained prominence in recent years, particularly in the domain of board games and decision-making under uncertainty. It is characterized by its ability to efficiently explore large search spaces by sampling potential future outcomes through Monte Carlo simulations. The algorithm iteratively builds a search tree by simulating random sequences of moves from the current game state and evaluating their outcomes through repeated simulations. By focusing computational resources on promising branches of the search tree, MCTS aims to guide the search towards regions of the game space that are more likely to lead to favorable outcomes. MCTS has demonstrated remarkable success in various domains, including games like Go, where traditional search algorithms struggle due to the game's immense complexity and branching factor.

**Tit-for-Tat** Axelrod (1981) is a simple but powerful strategy in the realm of repeated games and social dilemmas. The strategy is based on the principle of reciprocity, where an agent initially cooperates and then mimics the opponent's previous action in subsequent rounds. Specifically, Tit-for-Tat starts by cooperating in the first round and then replicates the opponent's last move in each subsequent round. Despite its simplicity, Tit-for-Tat has been shown to be remarkably effective in promoting cooperation and achieving favorable outcomes in various scenarios, including `Iterated Prisoner's Dilemma` and evolutionary simulations. Its success stems from its ability to balance cooperation and retaliation, fostering reciprocal behavior and encouraging cooperation among interacting agents.

## A4 LLM-vs-LLM Results

In Figure A6, we present the confusion matrix of NRA when various LLM agents are against GPT-3.5-turbo and GPT-4.

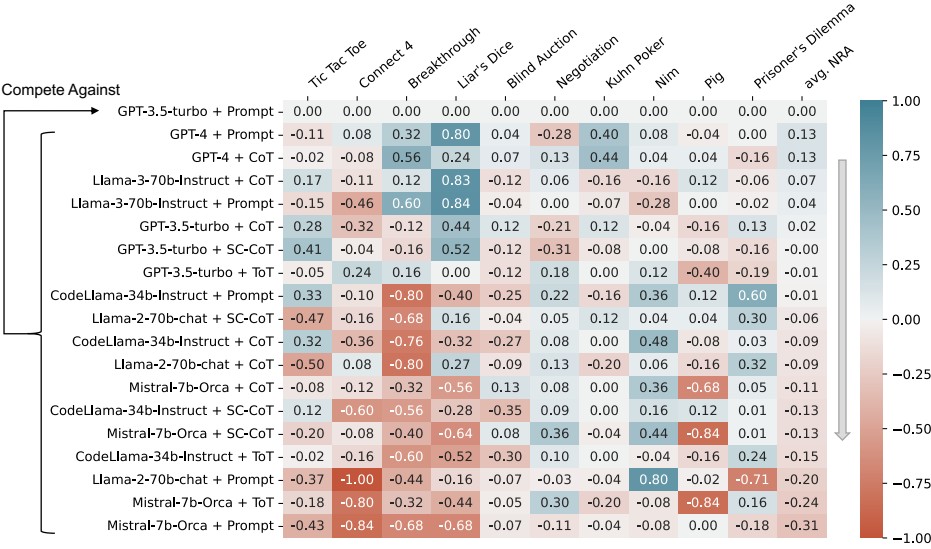

Figure A6: NRA confusion matrix of LLM vs. LLM across ten games ranked by average NRA. GPT-3.5-turbo with Prompt Agent serve as the common opponent against multiple combinations of LLMs with agents.

## A5 Prompt and Protocol

### A5.1 Modular Prompt Structure

When prompting LLMs to generate the next action during the course of a game, the prompt is composed of four individual components, to make sure all the participants access the same observations and information from environments:

**System Prompt** provides general guidance on how the LLMs should perform.

**Head Prompt** provides the general background and rules of the game.

**Observation Prompt** is formatted by a fixed game-wise template, providing sufficient observations from the environment regarding the current gaming state, to make LLMs capable of making decisions. The following provides the template used in the *Blind Auction* environment:

> Your budget is <VALUATION>. Your bid must be strictly lower than or equal to <VALUATION>. Your opponent also has an expected valuation and you do not know it.
> The legal actions are: <LEGAL_MOVES>.

Here <VALUATION> and <LEGAL_MOVES> are variables and are obtained from a unified <observation> object. In this way, all the participants are guaranteed to assess the same information.

**Reasoning Prompt** guides the LLM's generation process, e.g., "Let's think step by step" for the CoT Agent.

### A5.2 System Prompt

The system prompt is the initial text or context provided by the user to prompt the model to generate a response. This prompt serves as the starting point for the model to understand the user's query or input and generate a relevant and coherent response based on the provided context. We use the same system prompt for different tasks in our work. Our system prompt is presented as follows:

> **System Prompt:** You are a powerful gaming agent who can make proper decisions to beat the user in gaming tasks. You are a helpful assistant that strictly follows the user's instructions. You must answer your questions by choosing one of the legal moves given by the user!

### A5.3 Head Prompt

The head prompt is the text that explains the rules of the games to the LLMs. In our work, we designed different prompts for each of the games.

- **Tic-Tac-Toe**

> **Head Prompt:** Tic Tac Toe is a two-player game played on a grid. Players take turns marking a space with their respective symbols. The goal is to get 3 of one's own symbols in a row, either horizontally, vertically, or diagonally, before the opponent does. If all nine squares are filled and no player has three in a row, the game is a draw. The Tic Tac Toe game is played on a 3 by 3 grid, with the winning length as 3. Each move is represented by a string consisting of two parts: the column (C) and the row (R), in that order. For instance, C1R2 means the movement at the position of the first column and the second row of the grid. You are playing this game with the user (opponent).

- **Iterated Prisoner's Dilemma**

> **Head Prompt:** You and your partner are in the Prisoner's Dilemma situation. Specifically, if you <Testify> against your partner and your partner remains <Silent>, you will go free while your partner will get 3 years in prison on the main charge. If you remain <Silent> but your partner <Testify> against you, you will serve 3 years in prison and your partner will be set free. If you and your partner <Testify> against each other, you and your partner will each

serve 2 years. If both you and your partner remain <Silent>, you and your partner will each serve 1 year.

- **Breakthrough**

  **Head Prompt:** Breakthrough is a two-player game played on a rectangular board. Players take turns moving their pieces, which can move one space straight or diagonally forward if the target square is empty. A piece can also move diagonally forward to capture an opponent's piece. Capturing is optional, and a player can only capture one piece per turn. The goal is to be the first to reach the opponent's home row, the farthest row from the player. If all of a player's pieces are captured, they lose. The game does not allow draws, as pieces can only move forward or be captured. The Breakthrough board is identified by columns labeled starting from A (from left to right) and rows numbered 1 to 8 (from bottom to top). The intersection of a column and a row specifies a unique square on the board.

- **Connect Four**

  **Head Prompt:** Connect 4 is a two-player connection board game, where the players choose a color and then take turns dropping colored discs into a vertically suspended grid. The pieces fall straight down, occupying the next available space within the column. The objective of the game is to be the first to form a horizontal, vertical, or diagonal line of four of one's own discs. You are a gaming agent who aims to beat me in Connect 4 games. Each move is represented by a string consisting of two parts: the column (C) and the row (R), in that order. For instance, C1 means the first column.

- **First-price sealed-bid auction**

  **Head Prompt:** A first-price sealed-bid auction (FPSBA) is a common type of auction. It is also known as the blind auction. In this type of auction, all bidders simultaneously submit sealed bids so that no bidder knows the bid of any other participant. The highest bidder pays the price that was submitted.

  Each action is represented by $<x>$ where $x$ refers to the bid.

- **Kuhn Poker**

  **Head Prompt:** Kuhn poker is a simple model zero-sum two-player imperfect-information game, amenable to a complete game-theoretic analysis. In Kuhn poker, the deck includes only three playing cards: a King (K), a Queen (Q), and a Jack (J). One card is dealt to each player, and the third is put aside unseen. The players take turns either <Bet> to match the bet raised by the opponent or <Pass> to concede the game.

  If a player bets, the other player must either call the bet by matching it or fold by conceding the game. If both players pass, the game is over, and the player with the higher-ranking card wins. The card rankings are as follows: King (K) > Queen (Q) > Jack (J).

  You are playing Kuhn poker with the opponent. The actions are denoted by <Bet> and <Pass>.

- **Liar's Dice**

  **Head Prompt:** Liar's Dice is a game of bluffing and probability, played with two players and each player has 1 dice. During each turn, a player can either bid a higher quantity of any particular face value or the same quantity of a higher face value than the previous bid. Each player tries to outbid their opponent without being caught in a lie. The move in this game is denoted in $<x$ dices, $y$ value$>$, meaning there are at least $x$ dices with face values as $y$.

- **Pig**

  **Head Prompt:** Pig is a fast-paced dice game where players risk accumulating points with each roll but risk losing them all if they roll a 1. Each player must decide when to stop rolling and bank their points, aiming to be the first to reach 100 points. You are playing Pig with the other.

- **Nim**

> **Head Prompt:** In Nim, a strategic game with a set of four piles containing 1, 3, 5, and 7 matches respectively, players aim to avoid taking the last match. During each turn, a player may take any number of matches from a single pile, but must take at least one and cannot exceed the number remaining in that pile. The objective is to force the opponent to pick up the final match, thereby winning the game.
>
> The action is presented in $<$pile:$x$, take:$y>$, which means take $y$ match(es) from the $x$-th pile.

- **Negotiation**

> **Head Prompt:** You are negotiating the division of Peppers, Strawberries, and Cherries with the opponent. Different values these items hold for both you and your opponent. The process is structured into two stages per round: the proposal stage and the utterance stage.

### A5.4 Observations

Our research team has developed a range of observation prompts tailored to different types of games. The list of these prompts is presented below.

- **Tic-Tac-Toe**

> **Observation Prompt:** Your opponent has finished actions: $<$OPPONENT_MOVES$>$. You have finished actions: $<$SELF_MOVES$>$.

- **Iterated Prisoner's Dilemma**

> **Observation Prompt:** You have been through this situation in the past and here are the decisions you and your partner made: (In the $idx + 1$ th round, you decided to $<$MOVE$>$ and your opponent decided to $<$OPPONENT_MOVE$>$) * $n$ round

- **Breakthrough**

> **Observation Prompt:** The board now looks like : $<$BOARD_PREVIEW$>$. Among which, the letter 'b' represents a black piece, while the letter 'w' represents a white piece. And the character "." represents vacant space. The numbers in the board are the indexes of the rows. Your opponent has finished actions: $<$OPPONENT_MOVES$>$.You have finished actions: $<$SELF_MOVES$>$.

- **Connect Four**

> **Observation Prompt:** Your opponent has finished actions: $<$OPPONENT_MOVES$>$. You have finished actions: $<$SELF_MOVES$>$.

- **First-price sealed-bid auction**

> **Observation Prompt:** Now, you are in an auction with an opponent. You want to win the object and at the same time, your budget is $<$VALUATION$>$. Your bid must be strictly lower than or equal to $<$VALUATION$>$. You shall bid wisely against your opponent. Your opponent also has an expected valuation and you do not know it.

- **Kuhn Poker**

> **Observation Prompt:** In this match, your card is $<$CARD$>$. Here are the past moves in this match: $<$SELF_MOVES$>$, $<$OPPONENT_MOVES$>$.

- **Liar's Dice**

> **Observation Prompt:** Currently, the face value of your dice is $<$FACE_VALUE$>$. Last time, the opponent called $<$OPPONENT_LAST_ACTION$>$. You are playing the Liar's Dice with another opponent. Therefore, there are only two dice in total.

- **Pig**

> **Observation Prompt:** Right now, your current score is <AGENT_CURRENT_SCORE> and your opponent's current score is <OPPONENT_CURRENT_SCORE>. In this turn, you have earned <TURN_TOTAL_SCORE> score.

- **Nim**

> **Observation Prompt:** Currently, the 1st pile has <PILES[0]> match(es), the 2nd pile has <PILES[1]> match(es), the 3rd pile has <PILES[2]> match(es), 4th pile has <PILES[3]> match(es).

- **Negotiation** We proposed two different prompts for the "proposal" turn and "utterance" turn respectively.

For the "proposal" turn, we have:

> **Observation Prompt:** Now, the opponent propose to take <OPPONENT_PROPOSAL_TAKE[0]> peppers, <OPPONENT_PROPOSAL_TAKE[1]> strawberries, and <OPPONENT_PROPOSAL_TAKE[2]> cherries from the item pool. Last time, the utterance of the opponent was to take <OPPONENT_UTTERANCE_TAKE[0]> peppers, <OPPONENT_UTTERANCE_TAKE[1]> strawberries, and <OPPONENT_UTTERANCE_TAKE[2]> cherries from the item pool.
>
> Now, it is your decision. If you find the proposal raised by the opponent is acceptable, you should output Agree. Otherwise, you should output your proposal in the format <Proposal: [$a$, $b$, $c$]>.

For the "utterance" turn, we have:

> **Observation Prompt:** Last time, you propose to take <AGENT_PROPOSAL_TAKE[0]> peppers, <AGENT_PROPOSAL_TAKE[1]> strawberries, and <AGENT_PROPOSAL_TAKE[2]> cherries from the item pool. Last time, the utterance of the opponent was to take <OPPONENT_UTTERANCE_TAKE[0]> peppers, <OPPONENT_UTTERANCE_TAKE[1]> strawberries, and <OPPONENT_UTTERANCE_TAKE[2]> cherries from the item pool.
>
> Now, it is your turn to provide your utterance regarding the division of items. The utterance is what you want to tell your opponent and does not mean your real intent. You should output your utterance in the format <Utterance: [$a$, $b$, $c$]>.

### A5.5 Reasoning Prompt

- **Prompt agent:** Prompt agent does not necessitate the use of LLMs to apply any predetermined strategy prior to decision-making. Rather, it simply requests LLMs for inference and subsequently provides the outcome.

> You must choose a legal action to set up advantages. Your output must be in the following format:
> Action: Your action wrapped with <>, i.e., <format>
> Please return your answer without explanation!

- **CoT agent:** CoT agent makes LLMs consider the given observation first, then give out the action according to its thinking.

> First think about your current situation, then you must choose one action from legal actions to set up advantages.
> Your output must be in the following format strictly:
> Thought: Your thought.
> Action: Your action wrapped by <>, i.e., <format>
> Remember, you can only choose one move from the legal actions.

- **SC-CoT agent:** SC-CoT agent is an advanced version of the CoT agent. It obtains actions from multiple CoT trajectories. It employs the same prompt templates as in the CoT agent.

> First think about your current situation, then you must choose one action from legal actions to set up advantages.
>
> Your output must be in the following format strictly:
>
> Thought: Your thought.
>
> Action: Your action wrapped by $<>$, i.e., $<$format$>$
>
> Remember, you can only choose one move from the legal actions.

- **ToT agent:** we follow the text generation task implementation in the official codebase of ToT [16]. Specifically, the ToT is factorized into 1). candidate thought generation, 2). thought voting, 3). candidate action generation, 4). action voting:

Here we provide the basic prompt template used in ToT.

> **Step Prompt:** First think about your current situation, then choose one move from legal positions to set up advantages.
>
> Your output should be of the following format:
>
> Thought:
>
> Your thought.
>
> Move:
>
> Your action wrapped with $<>$, e.g., $<$format$>$

After executing step prompts in a breath-first search manner, we utilize the original ToT vote prompt:

> **Vote Prompt:** Given an instruction and several choices, decide which choice is most promising. Analyze each choice in detail, then conclude in the last line "The best choice is s", where s the integer id of the choice.

### A5.6 Sanity Check

To evaluate the effectiveness of our framework, we perform a sanity check by calculating the completion rates of each game. The completion rates are calculated as $\frac{50}{N}$ where $N$ is the number of matches that will take to achieve 50 valid matches. Here, a valid match means all the participants will always generate legal moves at each turn of the match. Results are summarized in Table A7. We show that all the LLM agents achieve $\geq 90\%$ completion rate, showing that the prompts are properly configured and LLMs are capable of following instructions to finish the game.

Table A7: Sanity check. The completion rates of LLM agents over all the games.

| Backbone LLM | Reasoning | Tic Tac Toe | Connect 4 | Breakthrough | Liar's Dice | Blind Auction | Negotiation | Kuhn Poker | Nim | Pig | Prisoner's Dilemma | avg |
|---|---|---|---|---|---|---|---|---|---|---|---|---|
| | Prompt | 100% | 100% | 98% | 98% | 100% | 100% | 100% | 100% | 100% | 100% | 100% |
| GPT-3.5-turbo | CoT | 100% | 100% | 98% | 98% | 100% | 100% | 100% | 98% | 100% | 100% | 100% |
| | SC-CoT | 100% | 100% | 100% | 98% | 100% | 100% | 100% | 100% | 100% | 100% | 100% |
| | Prompt | 100% | 100% | 100% | 100% | 100% | 100% | 100% | 100% | 100% | 100% | 100% |
| Llama-2-70b-chat | CoT | 81% | 98% | 64% | 100% | 89% | 69% | 100% | 100% | 100% | 98% | 90% |
| | SC-CoT | 89% | 91% | 81% | 100% | 94% | 68% | 100% | 100% | 100% | 100% | 92% |
| | Prompt | 98% | 100% | 89% | 100% | 100% | 100% | 100% | 100% | 100% | 100% | 99% |
| CodeLlama-34b-Instruct | CoT | 82% | 100% | 58% | 100% | 100% | 78% | 100% | 100% | 100% | 100% | 92% |
| | SC-CoT | 71% | 100% | 71% | 100% | 100% | 77% | 100% | 100% | 100% | 100% | 92% |
| | Prompt | 98% | 100% | 98% | 98% | 100% | 100% | 100% | 100% | 100% | 100% | 99% |
| Mistral-7b-Orca | CoT | 94% | 98% | 100% | 100% | 100% | 100% | 100% | 100% | 100% | 100% | 99% |
| | SC-CoT | 93% | 100% | 100% | 100% | 100% | 100% | 100% | 100% | 100% | 100% | 99% |

## A6 How Temperature Affects LLM Performance

To study how the temperature used in generating LLMs' responses affects performances, we conduct experiments by making LLMs with 0.2 temperature (the default setting as in our paper) play against LLMs with 0.4/0.6/0.8 temperature, over CodeLlama-34b-Instruct and GPT-3.5-turbo-1106. For each experiment, we run 20 matches. The reasoning method is the PromptAgent. The results are summarized as in Table A8. We show that a larger temperature will result in worse performance for deterministic games, while it has a model-specific effect for probabilistic games.

---

[16] https://github.com/princeton-nlp/tree-of-thought-llm/blob/master/src/tot/prompts/text.py

Table A8: The affect of various temperatures for generation sampling.

| Model | Temperature | avg. NRA in Probabilistic Games | avg. NRA in Deterministic Games |
|---|---|---|---|
| CodeLlama-34b-Instruct | 0.4 | -0.13 | -0.01 |
| CodeLlama-34b-Instruct | 0.6 | -0.16 | -0.05 |
| CodeLlama-34b-Instruct | 0.8 | -0.16 | -0.10 |
| GPT-35-turbo | 0.4 | 0.04 | -0.10 |
| GPT-35-turbo | 0.6 | 0.06 | -0.12 |
| GPT-35-turbo | 0.8 | 0.02 | -0.34 |

## A7 Elo Rating System

The Elo rating system Elo (1960) is a popular method for calculating the relative skill levels of players in two-player games such as Chess. It was used by various organizations to rank players. Assume there are two players: $A$ and $B$, and each player has a rating, $R_A$, $R_B$, which is a numerical value representing their skill level. The expected score for a player is the probability that the player will win against another player:

$$E_A = \frac{1}{1 + 10^{(R_B - R_A)/400}}$$

$$E_B = \frac{1}{1 + 10^{(R_A - R_B)/400}}.$$

After a match between $A$ and $B$, the real values, $S_A$ and $S_B$, are defined as

- If Player $A$ wins, $S_A = 1$ and $S_B = 0$
- If Player $B$ wins, $S_A = 0$ and $S_B = 1$
- If the game is a draw, $S_A = S_B = 0.5$

Then, the updated rating $R_A^{'}$ and $R_B^{'}$ are calculated as:

$$R_A^{'} = R_A + K * (S_A - E_A))$$

$$R_B^{'} = R_B + K * (S_B - E_B)),$$

where $K$ is a constant that determines how much the rating changes after a game. A higher $K$ results in a larger change. In our paper, the initial rating is set to 1500, i.e., $R_A = R_B = 1500$, and $K = 20$.

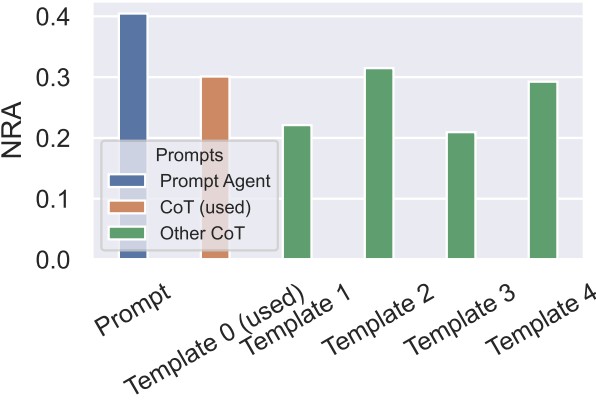

Figure A7: Investigating the sensitivity of Chain-of-Thought prompt. Prompt (used) and CoT (used) refer to the prompts utilized by the Prompt Agent and the CoT Agent in this paper. Results are obtained from the model GPT-3.5-turbo over all the game-theoretic tasks. Please refer to Table A9 for Template 0 to Template 4.

## A8  Chain-of-Thought Sensitivity

We provide five different CoT strategies over the GPT-3.5-turbo model as shown in Table A9 the results presented in Figure A7.

We also include instances as shown in Table A10 where CoT agents were unable to produce legal outcomes, which serves as evidence of their limitations.

Table A9: Different Chain-of-Thought strategies.

| Index | Chain-of-Thought Prompt |
|---|---|
| 0 | First, think about your current situation. |
| 1 | First, think about your current situation by thinking carefully, logically, and step by step. |
| 2 | First, provide your thoughts. |
| 3 | First, analyze and think about your current state and provide your strategy to beat your opponent. |
| 4 | First, analyze the state and provide your strategy to beat your opponent. |

Table A10: Fail cases in different Chain-of-Thought prompts.

| Index | Game | Chain-of-Thought Prompt |
|---|---|---|
| 0 | Breakthrough | Thought: I can see that my opponent has advanced their pieces quite far, and I need to make strategic moves to gain an advantage. I should prioritize moving my pieces forward while also considering potential capture opportunities. Action: <c4->b5> |
| 1 | Breakthrough | Thought: Considering the current board position and the opponent's recent moves, it seems that the best move would be to advance the white piece at b3 to c4, capturing the black piece at c4. This move would help in gaining control of the board and putting pressure on the opponent. Action: <c3->c4*> |
| 2 | Breakthrough | Thought: I see that the opponent's pieces are advancing towards my home row, and they have already captured one of my pieces. I need to start advancing my pieces and try to create opportunities for capturing the opponent's pieces to gain an advantage. Action: <b2->b3> |
| 3 | Nim | Thought:In this state, the total number of matches is 4, which is a losing position in Nim. To win, I need to force the total number of matches to be a multiple of 4 after my turn. This means I should aim to leave 3, 7, 11, or 15 matches for my opponent after my move. Action: <pile:4, take:3> |
| 4 | Nim | Thought: The current state of the game is such that the total number of matches remaining is 3, which is a losing position for the current player. To win, the strategy should be to force the opponent into a losing position by making a move that leaves 2 matches remaining. Action:<pile:4, take:6> |

## A9  Error Pattern Demonstrations

We present demonstrations of error patterns in Table A11.

Table A11: We conclude 5 common error patterns in GTBENCH, including misinterpretation, factual error, math calculation, and over-confidence. For each demonstration, the explanation explains why LLM generation is incorrect.

| Game | Category | Demonstrations |
|---|---|---|
| Breakthrough | Factual Error | **LLM Generation:** "Considering the current situation, moving the piece at b6 to either a5, b5, or c5 would allow me to advance further into the opponent's territory. However, this could potentially expose my piece to being captured. On the other hand, moving a piece from the back row (a8, b8, or c8) to b7 could potentially block the opponent's advance and provide a defensive line. Action:<b8->b7>". 

**Explanation:** In the third round of this match, moving the back row renders it impossible to hinder the opponent's forward movement. |
| Nim | Calculation Error | **State:** 3 matches in Pile 2, 5 matches in Pile 3, and 7 matches in Pile 4. 
**LLM Generation:** "The current state of the game is such that the 2nd pile has 3 matches, the 3rd pile has 5 matches, and the 4th pile has 7 matches. The goal is to avoid being the one to pick up the last match. To achieve this, I need to consider the binary representations of the pile sizes and aim to make the XOR of all pile sizes equal to zero. This will put me in a winning position. Currently, the XOR of all pile sizes is 4 (011 in binary). To make this zero, I need to remove 4 matches from the 3rd or 4th pile. Action:<pile:3, take:4>". 
**Explanation:** The XOR of the binary representations of 3, 5, 7 is 1, not 4. |
| Iterated Prisoner's Dilemma | Over-confidence | **LLM Generation:** "Given the history of the previous round, where both my partner and I chose to remain silent, it seems that we are both inclined towards cooperation. This is a positive sign, as it indicates a level of trust and mutual understanding. Considering the potential outcomes, the safest option is to remain silent. If my partner also remains silent, we both serve the least possible time. Action: <Silent>" 
**Explanation:** The player is overconfident that the partner will cooperate and keep silent. |
| Connect-4 | Misinterpretation | **LLM Generation:** "I have analyzed the current situation and identified that my opponent has taken the center column, which gives them a significant advantage. To counter this, I will take the corner column to create a potential winning line. Action: <C1>." 
**Explanation:** Currently, the center column is unoccupied. |
| Tic-Tac-Toe | Endgame Misdetection | **State:** The opponent finished actions <C2R1> and <C2R2>, the agent finished moves <C1R1> and <C3R1>. 
**LLM Generation:** I have one mark in the first column, and one in the third. I could potentially win if I get the chance to mark the third row of the first column in the next turn. Action: <C1R3> 
**Explanation:** This is an immediate loss situation because the opponent has two pieces connected in a row and <C2R3> is unoccupied. The agent missed that they should take action <C2R3> to avoid lose |

## A10 Regret Value

### A10.1 Regret Value for `Blind Auction`

Assume that the last round of bidding was $b_1$ for the player and $b_2$ for the opponent. Assume the player's valuation is $v$, then the regret value is calculated by

```
if b_1 > b_2 + 1:
    regret = b_1 - (b_2 + 1)
else:
    if (b_2 + 1) < v:
        regret = v - (b_2 + 1)
    else:
        regret = 0
```

### A10.2 Regret Value for `Iterated Prisoner's Dilemma`

The regret value `Iterated Prisoner's Dilemma` is simply the accumulation of the regret value of per-turn `Prisoner's Dilemma`:

```
if player_move == 'Testify' and opponent_move == 'Silent':
    regret = 0
elif player_move == 'Testify' and opponent_move == 'Testify':
    regret = 0
elif player_move == 'Silent' and opponent_move == 'Testify':
    regret = 1
else:
    regret = 2
```

## A11 User Interfaces of GTBench Leaderboard

The user interfaces of GTBENCH leaderboard are presented in Figures A8 and A9.

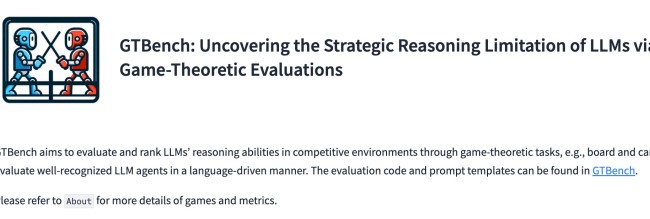

**GTBench: Uncovering the Strategic Reasoning Limitation of LLMs via Game-Theoretic Evaluations**

GTBench aims to evaluate and rank LLMs' reasoning abilities in competitive environments through game-theoretic tasks, e.g., board and card games. It utilizes 10 widely recognized games supported by OpenSpiel and evaluate well-recognized LLM agents in a language-driven manner. The evaluation code and prompt templates can be found in GTBench.

Please refer to `About` for more details of games and metrics.

The template is borrowed from Open LLM Leaderboard.

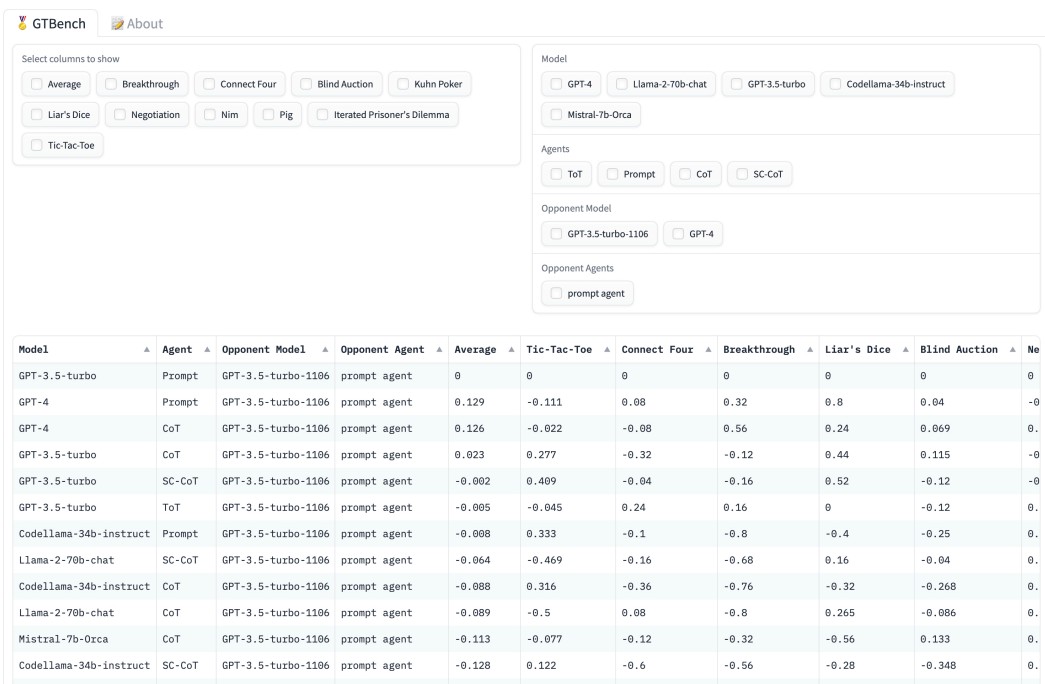

Figure A8: The user interface of GTBENCH leaderboard.

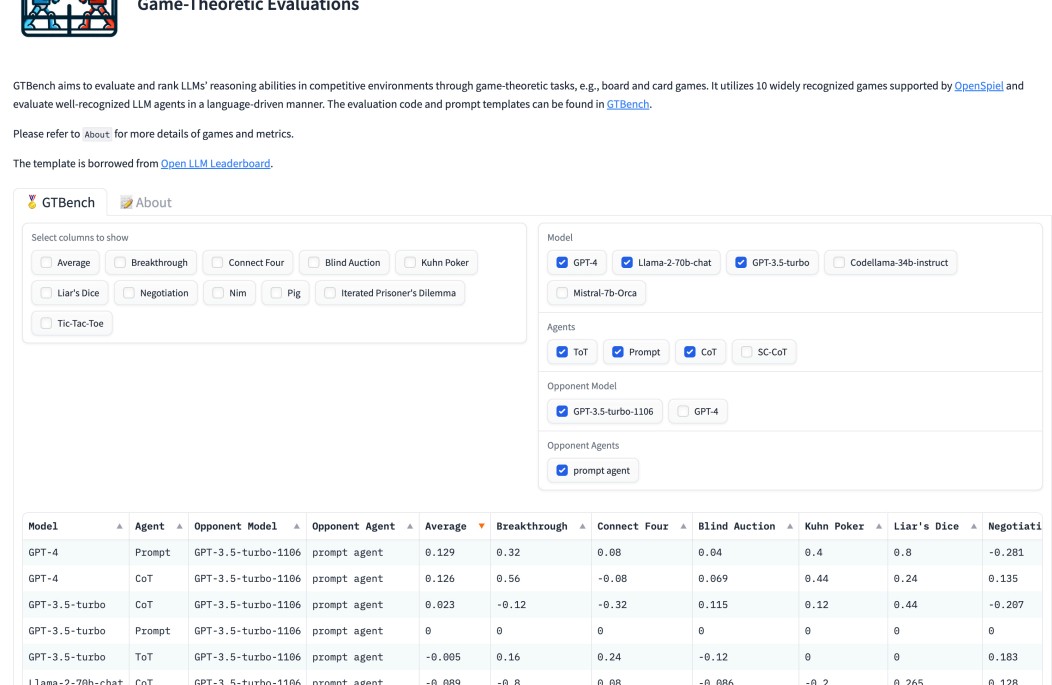

Figure A9: The user interface of GTBENCH leaderboard when various LLMs/agents and opponents are selected.

