# OpenReview forum: "GTBench: Uncovering the Strategic Reasoning Capabilities of LLMs via Game-Theoretic Evaluations"
_NeurIPS.cc/2024/Conference — NeurIPS 2024 poster_

### Official Review · Reviewer_L1Yq · 2024-07-05

**Soundness:** 3
**Presentation:** 3
**Contribution:** 1
**Rating:** 4
**Confidence:** 4

**Summary:**

This paper tries to evaluate the strategic reasoning abilities of LLM. Therefore, 10 games are chosen where LLMs is trying to solve the game. This paper includes various open- and closed-source LLMs into consideration and build a benchmark for easy evaluation.

**Strengths:**

Evaluating the strategic reasoning is important and the evaluation includes various LLMs into consideration.

**Weaknesses:**

The evaluation protocol is questionable. More comments and questions are in the following section.

**Questions:**

1. Does the evaluation really evaluate the strategic reasoning? Basically the evaluation is letting the LLM to play as one of the player in the game. However, this is much like a decision making problem, especially when the opponent is also a LLM-agent, where the LLM agent is largely stationary. Therefore, I would like to ask the authors provide the justification about why the evaluation is about strategic reasoning, not the decision making?

2. Also about the strategic reasoning. The selected games only focus on competitive zero-sum games. what about general-sum and multi-player games? Are cooperative games, e.g., hanabi, also requiring strategic reasoning? Even further, mixed cooperative and competitive, e.g., soccer, need the strategic reasoning? I think the strategic reasoning is not well-defined and fully discussed.

3. Does the evaluation really unlock the abilities of LLMs? The evaluation is focusing the prompting. However, for games, especially unfamiliar games for LLMs, exploration is important. Therefore, a memory or long in-context learning of the exploration experience should be included for the evaluation of strategic reasoning in games.

**Limitations:**

More limitations should be discussed.

---

> ### Author Rebuttal · Authors · 2024-08-07
>
> We thank you for your valuable and insightful comments!
>
> >Q1: Does the evaluation really evaluate the strategic reasoning? Basically the evaluation is letting the LLM to play as one of the player in the game. However, this is much like a decision making problem, especially when the opponent is also a LLM-agent, where the LLM agent is largely stationary. Therefore, I would like to ask the authors provide the justification about why the evaluation is about strategic reasoning, not the decision making?
>
> Conceptually, strategic reasoning and decision making are related yet distinct in their focus[1-2]: Decision making tends to be more (1) **immediate** and (2) **focused on selecting the best option among alternatives**, whereas strategic reasoning involves (1) **long-term planning** and (2) **anticipating others’ actions**. In Section 2.2, we explicitly describe the gameplay process as a Markov Decision Process, involving multiple players and multi-turn action execution by participants.
>
> Games in GTBench involve long-term planning and emphasize competition arising from others’ actions. Specifically, each game comprises multiple turns, with Breakthrough often taking over 20 turns per match. In each turn, the decision of an LLM agent is constrained by factors such as game rules, past moves from opponents, and the current game state. The agent must achieve long-term planning, such as constructing a **FORK** in board games which requires multiple turns, and maintain long-context reasoning, such as in Negotiation which involves reviewing past moves and actions, with participation from at least two players. From this perspective, GTBench aligns more closely with strategic reasoning than with immediate decision making. We agree that decision making is the foundation of strategic reasoning, and we will add further discussion on the scope of this paper in our next revision.
>
> Reference:
>
> [1]Kahneman, Daniel. Thinking, fast and slow. macmillan, 2011.
>
> [2]Dixit, Avinash K., and Barry Nalebuff. The art of strategy: a game theorist's guide to success in business & life. WW Norton & Company, 2008.
>
>
> >Q2: Also about the strategic reasoning. The selected games only focus on competitive zero-sum games. what about general-sum and multi-player games? Are cooperative games, e.g., hanabi, also requiring strategic reasoning? Even further, mixed cooperative and competitive, e.g., soccer, need the strategic reasoning? I think the strategic reasoning is not well-defined and fully discussed.
>
> We would like to highlight that GTBench does **NOT** solely focus on zero-sum games: the taxonomy of zero-sum versus general-sum is a major classification in GTBench (Table 1, column 2), and 4 out of the 10 games are general-sum, specifically Negotiation, Blind Auction, Pig, and Iterated Prisoner’s Dilemma.
> We agree that cooperative games and multi-player environments provide valuable evaluations for strategic reasoning. Two of our games, Negotiation and Iterated Prisoner’s Dilemma, already involve collaboration. The mentioned Hanabi is a purely cooperative environment, which poses challenges in identifying individual relative skills during gameplay. We will extend our GTBench to multi-player games. In fact, GTBench already supports it for some games. For instance, we implemented the game PIG with 3 players: GPT-3.5-turbo PromptAgent vs CoTAgent vs ToTAgent. We report the win rate of each player over 50 matches:
>
> |agent|PromptAgent|CoTAgent|ToTAgent|
> |---|---|---|---|
> |Win rate|46%|42%|12%|
>
> We would like to emphasize that LLM strategic reasoning evaluation is still in its early stages. This paper aims to evaluate the strategic reasoning of LLM agents in competitive environments. Experimental results indicate that LLM agents are largely ineffective in these simple competitive games. Therefore, GTBench could serve as a unified starting point for this research domain. Our GTBench will be maintained long-term and will support more diverse and complex strategic scenarios in the future.
>
> >Q3: Does the evaluation really unlock the abilities of LLMs? The evaluation is focusing the prompting. However, for games, especially unfamiliar games for LLMs, exploration is important. Therefore, a memory or long in-context learning of the exploration experience should be included for the evaluation of strategic reasoning in games.
>
> We acknowledge that designing advanced exploration mechanisms and unlocking the abilities of LLMs in strategic reasoning is impactful. However, our current paper **does not** aim to fully realize the potential of LLMs in these games. This is because a unified strategic reasoning benchmark is needed before delving into such designs. The scope of this paper is to (1) provide a unified evaluation framework and (2) benchmark common LLM reasoning agents in various game-theoretic scenarios. Designing advanced strategic reasoning agents and fully unlocking model capabilities will be our next step in this domain. It is also worth noting that the implemented Tree-of-Thought (ToT) agent [1] integrates exploration into both thought space and action space through thought decomposition and generation. However, ToT is less effective due to (1) limited exploration depth and (2) inaccurate reward estimation. This underscores that designing effective exploration mechanisms for LLM agents in strategic reasoning remains an open problem, requiring specific design considerations. We will address this in future work and discuss it further in our next revision.
>
> Reference:
>
> [1]Yao, Shunyu, et al. "Tree of thoughts: Deliberate problem solving with large language models." Advances in Neural Information Processing Systems 36 (2024).

---

> > ### Comment · Reviewer_L1Yq · 2024-08-10
> > **Thanks for the response**
> >
> > 1. About the definition of the decision making and strategic reasoning. I do not agree about the strategic reasoning. Generally, the strategic behavior is mostly investigated in game theory. What game theory differs from RL is that it need to simulate the play process of the two or more players, thus the fictitious play is proposed. In this paper, it is still decision making, not strategic reasoning. Surely this would be a definition problem, but I think this is important.
> >
> > 2. Sorry for the zero-sum competitive claim. but your response do not address my concerns. does the cooperative, mixed cooperative & competitive games also need strategic reasoning? ——based on your definition, cooperative game is definitely need to infer other's action. but is ignored in your paper.
> >
> > 3. I agree that benchmarking LLM in games still in its early stage. but I am just skeptical about the claim that benchmarking LLM in games = evaluating the strategic reasoning, also that considering one player's action is strategic reasoning.
> >
> > Given that my score is the border reject, which is not that negative, and the author's rebuttal do not convince me about the suitability of the methods, I would keep my score.

---

> > > ### Author Response · Authors · 2024-08-11
> > >
> > > We thank you for your prompt response.
> > >
> > > 1. In our last response, we explicitly mention that the major difference between decision making and strategic reasoning is the consideration of other players, i.e., “**(2) anticipating others’ actions**” (also described in Section 2.2). We would appreciate further clarification on how the proposed definition ("**simulate the play process of the two or more players**") differs from ours.
> > >
> > >
> > > 2. Cooperative games do require strategic reasoning. We already highlighted that collaboration is one of the major categories in the taxonomy of GTBench in Table 1. Games such as the Iterated Prisoner’s Dilemma require cooperative strategies to maximize system rewards (please refer to the definition of Prisoner's Dilemma **L575-576**, and the definition of Tit-for-Tat **L695-702** which both clearly mention cooperation). Thus, **we would like to politely disagree with the claim that we ignore cooperative games**.
> > >
> > > 3. In Lines 41-48, we mention that games provide rigorous rules and well-defined action/state spaces, involving multi-player interaction (section 2.2, the gameplay is denoted as Markov decision process), which is well-aligned to the goals of strategic reasoning. We appreciate further clarification on the concerns of evaluating strategic reasoning through games.  **Two-player games, i.e., considering one player’s action, are well-recognized as strategic reasoning in game theory literature [1-3]**. Besides, the reviewer has mentioned the definition of strategic reasoning as “simulate the play process of the two or more players”, indicating that two-player games are strategic reasoning. Also, we have provided the 3-player Pig experiments in our previous response to support that GTBench supports multi-player environments.
> > >
> > >
> > > Reference:
> > >
> > > [1] Hedden, Trey, and Jun Zhang. "What do you think I think you think?: Strategic reasoning in matrix games." Cognition 85.1 (2002): 1-36.
> > >
> > > [2] Abramson, Bruce. "Control strategies for two-player games." ACM Computing Surveys (CSUR) 21.2 (1989): 137-161.
> > >
> > > [3]Gutierrez, Julian, Paul Harrenstein, and Michael Wooldridge. "Expressiveness and complexity results for strategic reasoning." (2015).

---

### Official Review · Reviewer_PA1v · 2024-07-12

**Soundness:** 4
**Presentation:** 4
**Contribution:** 3
**Rating:** 7
**Confidence:** 4

**Summary:**

This paper proposes a benchmark for evaluating the strategic reasoning of LLMs. The benchmark includes ten games of various types. The authors use these games to conduct competitive experiments between LLMs and traditional methods, as well as LLM-vs.-LLM. The paper then analyzes the experimental results and model behavior, and examines the game-theoretic properties of LLMs.

**Strengths:**

1. The paper is logically clear, understandable, and well-written.
2. The experiments are comprehensive. The authors evaluate comparisons between LLMs and traditional methods and LLM-vs.-LLM competitions. They include multiple open-source and closed-source models and tests of various prompting methods.
3. The authors evaluate game-theoretic properties, including Nash equilibrium with regret and Pareto efficiency.

**Weaknesses:**

I didn't find any significant weaknesses, only a few questions.

**Questions:**

1. In Section 4.1, why does the tree-like prompting strategy ToT still lag significantly behind MCTS?
2. Is there any reference to classifying games in the benchmark? Why is it classified this way?
3. Why does the model perform better in probabilistic and dynamic games than in completely deterministic games? Is it that LLM performs better or that MCTS performs worse, making LLM appear better?

**Limitations:**

The authors have fully addressed the limitations in the paper.

---

> ### Author Rebuttal · Authors · 2024-08-07
>
> We thank you for your valuable and insightful comments.
>
> >Q1: In Section 4.1, why does the tree-like prompting strategy ToT still lag significantly behind MCTS?
>
> There are two potential reasons:
>
> 1. **Exploration Space**: MCTS has a significantly larger exploration space compared to ToT. In our experiments, MCTS is allowed to execute up to 1000 simulations to determine the next action. However, due to the complexity of the game and the cost of tokens, it is infeasible for ToT to traverse such a large number of simulations. For complex games like Breakthrough, completing even one full game simulation is impossible for ToT.
>
> 2. **Reward Backpropagation**: MCTS uses the actual win/loss signal obtained from simulations as a reward to determine which action should be selected. In contrast, ToT relies on LLMs voting or grading the actions, which is less accurate compared to the reward mechanism used by MCTS.
>
> >Q2: Is there any reference to classifying games in the benchmark? Why is it classified this way?
>
> We primarily classify games based on their game-theoretic properties, such as complete versus incomplete information, dynamic versus static, and probabilistic versus deterministic scenarios, which are fundamental and widely recognized in the field of game theory [1-3]. After categorizing the games according to these properties, we then identified and summarized the most commonly preferred abilities (Table 1).
>
> Reference:
>
> [1]Fraser, Niall. Conflict analysis. Ed. Keith W. Hipel. North-Holland, 1990.
>
> [2]Osborne, Martin J. An introduction to game theory. Vol. 3. No. 3. New York: Oxford university press, 2004.
>
> [3]Lanctot, Marc, et al. "OpenSpiel: A framework for reinforcement learning in games." arXiv preprint arXiv:1908.09453 (2019).
>
> >Q3: Why does the model perform better in probabilistic and dynamic games than in completely deterministic games? Is it that LLM performs better or that MCTS performs worse, making LLM appear better?
>
> Completely deterministic games are those in which all players have full knowledge of the game's state and the actions of other players. In this case, search-based solvers such as MCTS exhaustively explore possible future states and use statistical sampling to make highly informed decisions based on full information. Thus, MCTS approaches near-optimal performance in complete games, which significantly outperforms LLM agents. However, some information is hidden or unknown to the players (such as dice values or poker cards) in incomplete and probabilistic games, making MCTS simulation unavailable due to the hidden full-state information or uncertainty.
>
> We agree that MCTS may not be optimal for probabilistic games such as poker games. We further implemented the well-known **Counterfactual (CFR) solver** that is proven to be effective in finding Nash equilibria for in-complete games [1-3]. Then we conduct MCTS-vs-CFR experiments in 100 matches on the Kuhn Poker environment. The win rate of the CFR solver is 54%, which shows a slight advantage compared to the MCTS solver. We then re-run GPT-4 experiments when playing against the CFR solver. We observe that GPT-4 w/ Prompt achieves 0.33 NRA when against CFR, indicating that the conclusions presented in our paper are still consistent. We will include more CFR results in our next revision.
>
> Reference:
>
> [1]Zinkevich, Martin, et al. "Regret minimization in games with incomplete information." Advances in neural information processing systems 20 (2007).
>
> [2]Tammelin, Oskari, et al. "Solving heads-up limit texas hold'em." Twenty-fourth international joint conference on artificial intelligence. 2015.
>
> [3]Moravčík, Matej, et al. "Deepstack: Expert-level artificial intelligence in heads-up no-limit poker." Science 356.6337 (2017): 508-513.

---

> > ### Comment · Reviewer_PA1v · 2024-08-11
> > **Reply to the Rebuttal by Authors**
> >
> > Thanks for your response. I think it's an appropriate score.

---

### Official Review · Reviewer_HsvL · 2024-07-13

**Soundness:** 3
**Presentation:** 4
**Contribution:** 3
**Rating:** 7
**Confidence:** 4

**Summary:**

The paper proposes a benchmark to understand the strategic reasoning capabilities of llms. The authors present a suite of game theoretic tasks with different structures to do this. They use different evaluation metrics like ELOs and Relative advantage to compare different llms and prompting methods.

**Strengths:**

- The paper is clearly written and well motivated. It provides some structure to the growing literature of strategic reasoning with llms.
- A wide range of closed source, open source models are tested. A good set of prompts are used to test the models too!
- I particularly liked table 1 and the selection of different tasks with different characteristics.
- The normalized relative advantage is a good, interpretable metric
- The framework and taxonomy are clear and easy to understand.
- Section 4.4 gave some good insight into the types of errors made by llms
- I also liked reading the analysis in section 4.3, in particular that code pretraining helps with strategic reasoning.

**Weaknesses:**

- Characterizing human performance would strengthen the paper
- Including some qualitative reasoning traces of successes and failures might be insightful.
- Minor: This paper would be an ideal fit for the datasets and benchmarks track, instead of the main track. I dont think it should be penalized for this though!

Typos

Line 79: Characterize

Line 171: dynamic gaming → dynamic game

**Questions:**

See weaknesses.

**Limitations:**

The authors do a good job of addressing limitations.

---

> ### Author Rebuttal · Authors · 2024-08-07
>
> We thank you for your valuable and insightful comments!
>
> >W1: Characterizing human performance would strengthen the paper
>
> We provide a preliminary human opponent evaluation. Specifically, we selected 5 games from GTBench and organized matches with 5 graduate students. These participants are familiar with the rules but are not trained experts. Each human competed against MCTS and GPT-3.5-turbo in 10 matches for each game. We report the average NRA across these human participants.
>
> | setting | Tic-Tac-Toe | Connect-4 | Kuhn Poker | Liar’s Dice | Blind Auction |
> | --- | --- | --- | --- | --- | --- |
> |Human vs MCTS|-0.25|-0.55|0.09|-0.60|0.15|
> |MCTS vs Human |0.25 |0.55 | -0.09 | 0.60 | -0.15 |
> |GPT-3.5-turbo vs Human | -0.55 | -0.60 | -0.53 | -0.88 | -0.46 |
> |Human vs GPT-3.5-turbo | 0.55 | 0.60 | 0.53 | 0.88 | 0.46|
>
> Overall, our results indicate that even humans with moderate knowledge of these games outperform LLM agents significantly in strategic reasoning, in both complete and incomplete information scenarios. This underscores the limitations of current LLMs in strategic reasoning and highlights the need for improvement in this area. We will include these results and provide more detailed human-as-opponent evaluations in our next revision.
>
> >W2: Including some qualitative reasoning traces of successes and failures might be insightful.
>
> We have identified five common error patterns and documented the corresponding failure cases in Table A11 of our manuscript to provide a qualitative analysis. Here, we present a pair of success and failure logs from the Tic-Tac-Toe environment, with detailed annotations explaining the intent behind each step (the intent is derived from the CoT reasoning):
>
> **Success Traces (GPT-4-turbo w/ CoT Agent  beats Human)**
>
> **Step 1 (CoT Agent)**:
> Thought: As the first player, I have the advantage of making the first move. The center square is the most strategic position to start with, as it provides the most opportunities for creating a line of three. Action:<C2R2>
>
> **Step 2 (Human)**:<C3R2>
>
> **Step 3 (CoT Agent)**: Thought: Looking at the current state of the game, I can see that the center and one of the side squares are occupied. To maximize my chances of winning, I should aim to control one of the corners. This will give me multiple ways to create a line of three. Considering the available moves, I can choose either <C1R1> or <C3R1> to achieve this.Action:<C1R1>
>
> **Step 4 (Human)**: <C3R3>
>
> **Step 5 (CoT Agent)**: Thought: Looking at the current state of the game, my opponent has two marks in the third column. If they get another turn, they might place their mark in the first row of the third column and win the game. Therefore, I need to block this potential winning move. Action:<C3R1>
>
> **Step 6 (Human)**: <C2R1>
>
> **Step 7 (CoT Agent)**: Thought: Looking at the current state of the game, I notice that the opponent has two of their symbols in the third column. If they get a chance to place their symbol in the third row of the first column, they will win the game. Therefore, I need to block this potential winning move by placing my symbol in the third row of the first column. Action:<C1R3>
>
> **CoT Agent wins!** In this match, the CoT agent successfully blocked the human's connected pieces and identified its own winning move.
>
> **Failure Traces (Human Beats GPT-4-turbo w/ CoT Agent)**
>
> **Step 1 (Human)**: <C2R2>
>
> **Step 2 (CoT Agent)**: Thought:The opponent has taken the central position on the board, which is a strategic advantage because it opens up the most possibilities for creating a line of three symbols. To counter this, I should aim to block potential winning moves and set myself up for multiple winning paths. Placing my symbol in a corner can be a good strategy because it gives me two potential ways to create a line of three symbols (**NOTE: correct and popular strategy**). Among the corners, choosing <C1R1> or <C3R3> seems most advantageous as it does not immediately threaten by the opponent's current position but sets up a potential win in multiple directions. Action:<C1R1>
>
> **Step 3 (Human)**: <C2R1>
>
> **Step 4 (CoT Agent)**: Thought: Given the current state of the game, my opponent has placed their symbols in C2R2 and C2R1, forming a potential line vertically in column 2. My symbol is in C1R1. To maximize my chances of winning or at least blocking my opponent's path to victory, I should consider creating a two-front threat that forces my opponent to defend rather than attack (**NOTE: A hallucination occurred here; the opponent already has two connected pieces, so the first priority should have been blocking the opponent**). Placing my symbol in C3R3 would start to form a diagonal line from C1R1 to C3R3, and it also positions me to potentially create a vertical line in column 3. This move doesn't immediately block my opponent's potential win but sets up a strategic position for future moves.Action:<C3R3>
>
> **Step 5 (Human)**: <C2R3>
>
> **Human Wins!** The CoT agent lost because it failed to correctly prioritize its actions. With the opponent already having two connected pieces and no immediate winning move available for the CoT agent, the primary focus should have been on blocking the opponent.
>
> We will add more qualitative results in our next revision.

---

> > ### Comment · Reviewer_HsvL · 2024-08-11
> >
> > Thank you for your response! I'll keep my current score!

---

### Official Review · Reviewer_VyBq · 2024-07-13

**Soundness:** 3
**Presentation:** 3
**Contribution:** 3
**Rating:** 6
**Confidence:** 4

**Summary:**

This paper introduces GTBench, a set of 10 different games to test how well large language models can think strategically. The author found that while LLMs struggle with complete and deterministic games like Tic-Tac-Toe and Connect-4, they perform better in incomplete uncertain games like poker and negotiation. Code-pretraining improves their strategic thinking abilities. However, advanced thinking methods like Chain-of-Thought and Tree-of-Thought don’t always help and can sometimes make things worse. The latest open-source models, like Llama-3, are getting closer in performance to commercial models like GPT-4. Common mistakes LLMs make include misunderstanding game rules, being over-confident, and making calculation errors.

**Strengths:**

1. The paper is well-written and easy to understand.
2. The problem of evaluating LLMs' strategic reasoning abilities is meaningful. Creating such a benchmark is valuable for the research community.
3. The paper provides a detailed evaluation of LLMs across different game tasks. These tasks indeed measure the strategic reasoning of LLMs, even if some models already understand the optimal algorithms for those games. (For example, you could ask GPT-4 about the optimal strategy for some of these games, and it knows the optimal algorithm.)
4. The authors conducted extensive experiments using various base models, including reasoning methods like ToT and CoT. They had some interesting findings and analysis (concluded in the summary).

**Weaknesses:**

1. The paper claims that measuring strategic reasoning capabilities with games is missing in existing benchmarks. However, there are other benchmarks, such as MAgIC released last year, that consider benchmarking LLMs' strategic behavior using games. While there are differences, this weakens the claim of novelty.
2. Some of the selected games, like Tic-Tac-Toe, have known optimal strategies and are not complex enough. These games might not fully challenge the advanced strategic reasoning capabilities of LLMs. Even though the current evaluation is useful, as a benchmark intended for future use, it should be capable of evaluating more advanced or adapted LLM agents.
3. The benchmark focuses on a set of 10 games. It’s unclear how well the findings generalize to other strategic scenarios, even similar types of tasks. The results appear to be quite case-by-case. A broader range of tasks and scalable evaluation frameworks would make the benchmark more comprehensive.
4. The experiments primarily involve LLMs and traditional solvers. There is a lack of evaluation against human opponents, which could provide more insights into the models' performance in real-world strategic interactions. As a benchmark, I also expect to have other opponents (for example, the optimal algorithm, the RL based agent).

**Questions:**

Could you address the weakness 1, and try to discuss weakness 2-4?

**Limitations:**

If LLMs are trained on biased data, they might reinforce existing biases in strategic decision-making. Testing the LLM with different personas could show if this changes the game results and reveal potential biases.

---

> ### Author Rebuttal · Authors · 2024-08-07
>
> We thank you for your valuable and insightful comments!
>
> >W1: The paper claims that measuring strategic reasoning capabilities with games is missing in existing benchmarks. However, there are other benchmarks, such as MAgIC released last year.
>
> In Line 119, we meant to convey that some of the games and the taxonomy introduced in GTBench are not used by current benchmarks for evaluating LLM strategic reasoning. We recognize existing benchmarks that evaluate LLMs through games, such as Clemency, LMRL-Gym, and LLMArena, discussed in Section 2.1. While MAgIC focuses on the complex social and cognitive dimensions of LLMs through various games, our focus is on pure logical game-theoretic evaluation, which could complement our work. We will add more discussion about MAgIC and revise our statement in the next revision.
>
> >W2: Some games have known optimal strategies that might not fully challenge the capabilities of LLMs. A benchmark intended for future use should be capable of evaluating more advanced or adapted LLM agents.
>
> We would like to mention that only Tic-Tac-Toe and Connect-4 are solved games, meaning they have optimal strategies, while other games in GTBench remain complex and challenging without known optimal strategies. We are aware that some strategies for poker games are "approximately" optimal, but these strategies incorporate bluffing, which introduces uncertainty and prevents them from being universally optimal.
>
> Additionally, we would like to emphasize the following points:
>
> 1. GTBench includes both simple games like Tic-Tac-Toe and complex ones like Breakthrough, with up to 48 actions per turn. These complex games, popular in strategy competitions, pose a significant challenge for LLMs already struggling with simpler games.
>
> 2. LLM agents, even with optimal strategies, may still produce errors like hallucinations. Thus, comparing performance using statistics like the number of draws and deviations from the optimal strategy is valuable.
>
> 3. GTBench is flexible and extensible, with modular game implementations independent of LLM agent design. It will be maintained long-term, supporting more complex games and advanced agents in the future.
>
> We believe that the proposed GTBench will continue to be valuable for advancing LLM research. We will incorporate this discussion in our next revision.
>
> >W3: It’s unclear the generalization of findings to other scenarios. A broader range of tasks and scalable evaluation frameworks would make the benchmark more comprehensive.
>
> We would like to mention that GTBench focuses on gameplay and includes various strategic scenarios like competition, negotiation, and collaboration (Table 1). Our conclusions are based on general trends across all scenarios, not focus on specific types. For example, we compare the NRA of code-pretrained LLMs and chat LLMs across all scenarios to assess the effect of code pretraining, providing a broad overview of LLM agent performances.
>
> To address scalability, we offer adjustable action/state spaces and complexity, such as varying board sizes for Breakthrough and player numbers and winscore for PIG. As complexity increases, the advantage of powerful LLM agents like GPT-4 over GPT-3.5-turbo decreases, but our overall conclusions remain valid. Here are related results (GPT-3.5-turbo PromptAgent vs MCTS and GPT-4, reporting NRA/win rate over 50 matches):
>
> |Breakthrough|Column=3 (default) | Column=4|
> | --- | --- | --- |
> |Gpt-3.5-turbo vs MCTS |-1 |-1|
> |Gpt-4 vs GPT-3.5-turbo |0.32 |0.26|
>
> |3-Player Pig (GPT-3.5-turbo)|PromptAgent |CoTAgent |ToTAgent|
> | --- | --- | --- | --- |
> |Win rate | 46% | 42% | 12% |
>
> |Pig |Winscore=20 (default) |Winscore=30|
> | --- | --- | --- |
> |Gpt-3.5-turbo vs MCTS |-0.44 |-0.40|
> |Gpt-4 vs GPT-3.5-turbo |-0.04 |-0.06|
>
> We conclude that the relative advantage between LLM agents diminishes as games become more challenging. For example, increasing the board size in Breakthrough from 3 to 4 reduces GPT-4's NRA from 0.32 to 0.26, though it still outperforms GPT-3.5-turbo. This occurs because participants tend to show similar skill levels in more complex games. We will explore more extensible settings and support diverse games in future revisions.
>
>
> >W4: There is a lack of evaluation against human opponents and other opponents (for example, the optimal algorithm, the RL based agent).
>
>
> We conducted a preliminary human opponent evaluation by selecting 5 games (2 deterministic and 3 probabilistic) from GTBench. Five graduate students who are familiar with the rules but not experts in these games, played 10 matches against MCTS and GPT-3.5-turbo for each game:
>
> | setting | Tic-Tac-Toe | Connect-4 | Kuhn Poker | Liar’s Dice | Blind Auction |
> | --- | --- | --- | --- | --- | --- |
> |Human vs MCTS|-0.25|-0.55|0.09|-0.60|0.15|
> |MCTS vs Human |0.25 |0.55 | -0.09 | 0.60 | -0.15 |
> |GPT-3.5-turbo vs Human | -0.55 | -0.60 | -0.53 | -0.88 | -0.46 |
> |Human vs GPT-3.5-turbo | 0.55 | 0.60 | 0.53 | 0.88 | 0.46|
>
> The average NRA across participants showed that humans with moderate knowledge significantly outperform LLM agents in strategic reasoning in both complete and incomplete information scenarios. This highlights the limitations of current LLMs and the need for improvement. We will include these results and provide more detailed evaluations in our next revision.
>
> We agree that MCTS may not be optimal for probabilistic games like poker. We implemented the **Counterfactual Regret Minimization (CFR) solver**, known for finding Nash equilibria in incomplete games[1]. In 100 matches of Kuhn Poker, the CFR solver had a 54% win rate over MCTS, showing a slight advantage. We re-ran GPT-4 experiments against CFR, and GPT-4 with Prompt achieved a 0.33 NRA, consistent with our paper's conclusions. We will include more solvers and these results in our next revision.
>
> Reference:
>
> [1]Zinkevich, Martin, et al. "Regret minimization in games with incomplete information." Advances in neural information processing systems 20 (2007).

---

> > ### Comment · Reviewer_VyBq · 2024-08-14
> >
> > I appreciate the author's effort and the response to my comments. I was hoping for more solid work to give a higher score (like 7 or above). For example, adding more games, comparing with more solvers, or including some additional insights (perhaps thinking out-of-the-box). I understand that it’s hard to make big changes in a short time. I have increased my score and lowered my confidence.

---

> > > ### Author Response · Authors · 2024-08-14
> > >
> > > Thank you for your valuable feedback and support. We will incorporate additional results and details in our next revision.

---

> ### Author Response · Authors · 2024-08-13
> **We thank you for your valuable comments!**
>
> Dear reviewer VyBq,
>
> We would greatly appreciate your esteemed feedback, as this could be our last chance to resolve any outstanding issues or inquiries you might have. Should any elements of our work need additional explanation, we kindly ask you to inform us. We look forward to your valuable input and the chance to have a productive discussion to enhance our submission. Thank you!

---

### Author Rebuttal · Authors · 2024-08-07

## General Response

We appreciate all the valuable comments from the reviewers. We are pleased to know that our work is considered meaningful (Reviewer **VyBq**, **L1Yq**), valuable (Reviewer **VyBq**), comprehensive (Reviewer **VyBq**, **PA1v**), and insightful (Reviewer **PA1v**, **HsvL**).

Here are the major changes we have made:

1. We have added Human-as-Opponent experiments for further analysis (Reviewer **VyBq**, **HsvL**).
2. We have included the Counterfactual Regret Minimization (CFR) solver as a more powerful opponent in probabilistic games (Reviewer **VyBq**, **PA1v**).
3. We have provided detailed success/failure traces for qualitative analysis (Reviewer **HsvL**).
4. We have introduced extensible experiments for scalability, such as variations in board size, multi-player settings, and different gameplay win scores (Reviewer **VyBq**, **L1Yq**).
5. We have clarified the scope and the goal of this paper.

Further details are available in our individual responses. We are also open to providing additional clarifications and addressing any other concerns the reviewers may have.

---

### Decision · Program_Chairs · 2024-09-25

**Decision:**

Accept (poster)

**Comment:**

The paper introduces a benchmark designed to evaluate the strategic reasoning capabilities of LLMs using a suite of ten diverse games. These games range from deterministic ones like Tic-Tac-Toe and Connect-4 to uncertain, incomplete information games like poker and negotiation. The study finds that LLMs generally struggle with deterministic games but perform better in uncertain scenarios. Code-pretraining enhances their strategic thinking, though advanced methods like Chain-of-Thought and Tree-of-Thought can be counterproductive in some cases. The paper highlights that while open-source models like Llama-3 are closing the gap with commercial models like GPT-4, common errors persist, such as misunderstanding game rules, overconfidence, and calculation mistakes. The benchmark also facilitates competitive experiments between LLMs and traditional methods, as well as LLM-vs.-LLM, providing a comprehensive analysis of the game-theoretic properties of these models.

In general, reviewers were relatively positive about the benchmark, although it would have been nice (as always) to see a larger number of games and solvers. Nonetheless, the games selected provide a good range over categories (such as collaborative vs competitive, perfect vs hidden information, etc). The main point of concern is around the central claim as to whether the benchmark does in fact probe the ability of LLMs to perform strategic reasoning or whether this is simply a case of decision making.  Certainly the title is slightly misleading, in that the benchmark merely serves to evaluate, as opposed to "unlock" strategic reasoning abilities (which hopefully work down the line will do). Nonetheless, I think that the current benchmark is useful and intersting enough to be of utility to the field.